# Staggered Environment Resets Improve Massively Parallel On-Policy Reinforcement Learning

**Sid Bharthulwar**
Harvard University
sbharthulwar@college.harvard.edu

**Stone Tao**
UC San Diego
stao@ucsd.edu

**Hao Su**
UC San Diego
hao@sudo.ai

## Abstract

Massively parallel GPU simulation environments have accelerated reinforcement learning (RL) research by enabling fast data collection for on-policy RL algorithms like Proximal Policy Optimization (PPO). To maximize throughput, it is common to use short rollouts per policy update, increasing the update-to-data (UTD) ratio. However, we find that, in this setting, standard synchronous resets introduce harmful nonstationarity, skewing the learning signal and destabilizing training. We introduce staggered resets, a simple yet effective technique where environments are initialized and reset at varied points within the task horizon. This yields training batches with greater temporal diversity, reducing the nonstationarity induced by synchronized rollouts. We characterize dimensions along which RL environments can benefit significantly from staggered resets through illustrative toy environments. We then apply this technique to challenging high-dimensional robotics environments, achieving significantly higher sample efficiency, faster wall-clock convergence, and stronger final performance. Finally, this technique scales better with more parallel environments compared to naive synchronized rollouts.

## 1 Introduction

Reinforcement Learning (RL) has emerged as a powerful paradigm for tackling complex sequential decision-making problems, particularly in continuous control domains like robotics [8, 11, 10]. However, RL often depends on vast quantities of interaction data, a requirement that can be prohibitively expensive or slow to acquire in real-world settings. Massively parallel simulation environments, especially those accelerated on GPUs [14, 3, 20, 16], have enabled data-generation throughput on orders of magnitude greater than traditional CPU-based setups. The increase in data throughput has enabled far faster training of robotics models with successful sim2real deployments of locomotion [21, 15], state-based manipulation [5, 13], and vision-based manipulation [26, 32, 24].

Despite this paradigm shift in data generation capabilities, the core algorithms, particularly on-policy methods like Proximal Policy Optimization (PPO) [22], have often been adapted with only superficial changes—typically larger batch sizes and shorter per-environment rollouts ($K$) to increase the update-to-data (UTD) ratio [21]. This strategy, while seemingly maximizing hardware utilization, overlooks a critical interaction between the data collection process and the learning algorithm's stability when the task horizon ($H$) significantly exceeds the rollout length ($K \ll H$).

We argue that stable and efficient learning in this massively parallel regime requires more than just algorithmic re-tuning; it necessitates a modification to the environment interaction protocol itself.

39th Conference on Neural Information Processing Systems (NeurIPS 2025).

We introduce **staggered resets**, a simple yet highly effective technique that breaks this harmful synchronicity. By initializing parallel environments at diverse effective time steps distributed across the task horizon $H$, staggered resets ensure that each training batch contains a rich, temporally heterogeneous mix of experiences. This provides the learner with a more stationary and representative view of the overall task dynamics within every gradient update. To summarize our contributions:

- We precisely identify, evidence, and formulate the problem of cyclical batch nonstationarity stemming from synchronous full-episode resets combined with short rollouts ($K \ll H$) in massively parallel on-policy RL, explaining its detrimental impact on learning dynamics.

- We propose staggered resets, an elegant and easily implementable mechanism independent of the RL algorithm itself to ensure temporal diversity within training batches by desynchronizing the effective starting points of parallel environments across the task horizon.

- Through illustrative toy environments, we characterize the conditions under which this nonstationarity is most severe and staggered resets offer maximal benefit.

- We provide compelling empirical evidence on challenging, high-dimensional robotics tasks, demonstrating that staggered resets significantly improve sample efficiency, wall-clock convergence speed, final policy performance, and scalability with increasing parallelism compared to standard synchronous reset protocols.

## 2 Related Work

**Massively Parallel RL**    The inherent sample inefficiency of many reinforcement learning algorithms has significant research into scaling and parallelization to improve training time and performance. Early approaches, such as IMPALA and others [17, 2, 18, 6], utilized multiple CPU workers to achieve parallelism and scalability. These typically relied on benchmarks [31, 4, 28] built on top of CPU-based simulators like MuJoCo [27], PyBullet [1], PhysX etc. More recently, GPU-accelerated simulators [14, 3] and other JAX-based GPU-accelerated environments [20, 9] have enabled a much greater degree of parallelism, resulting in considerable speedups for training complex policies. Recent work has sought to replace the popular choice of PPO as an RL algorithm by improving scalability [12, 25]. In contrast our work is algorithm-agnostic and addresses the challenge of handling non-stationary data in synchronous highly-parallel regimes for on-policy algorithms like PPO.

**Nonstationarity in RL**    Nonstationarity in the data distribution is a recognized challenge within reinforcement learning. Such nonstationarity can stem from various sources, including changes in environment dynamics, the reward function, or, as pertinent to our work, the data collection process itself. Previous research has explored related issues such as catastrophic forgetting in continual learning scenarios [7] and representation collapse arising from biased data. The "primacy bias," wherein early experiences exert a disproportionate influence on RL training, has also been documented [19, 23]. Our work specifically highlights and aims to mitigate the cyclical nonstationarity induced by the interplay of synchronous resets and short rollouts in massively parallel RL settings.

**Short Rollouts in Parallel RL**    The use of short rollouts ($K \ll H$) in parallel RL is frequently motivated by the desire to increase update frequency (achieving a high update-to-data ratio) and maximize wall-clock training speed [30, 12]. While this strategy can be effective, underlying issues related to data distribution bias are often overlooked. Some implementations incorporate partial resets—resetting environments upon task success, failure, or termination—which can introduce some eventual desynchronization. However, this process can be slow and may prove insufficient to counteract the initial bias, particularly when $K$ is very small. Furthermore, while some simulation environment implementations include versions of the reset staggering technique we propose [21], these are typically not detailed in accompanying publications. To our knowledge, our work is the first to thoroughly investigate and analyze this method, as well as to characterize the environmental conditions under which it proves most effective.

# 3 Methodology

## 3.1 Preliminaries: PPO in Massively Parallel Environments

We consider the standard RL setting where an agent interacts with an environment modeled as a Markov Decision Process (MDP), defined by $(S, A, P, r, \rho_0, \gamma, H)$, where $S$ is the state space, $A$ is the action space, $P(s'|s, a)$ is the transition probability function, $r(s, a)$ is the reward function, $\rho_0$ is the initial state distribution, $\gamma \in [0, 1)$ is the discount factor, and $H$ is the maximum episode horizon. The goal is to learn a policy $\pi_\theta(a|s)$, parameterized by $\theta$, that maximizes the expected discounted cumulative return:

$$J(\pi_\theta) = \mathbb{E}_{s_0 \sim \rho_0, a_t \sim \pi_\theta(\cdot|s_t), s_{t+1} \sim P(\cdot|s_t, a_t)} \left[ \sum_{t=0}^{H-1} \gamma^t r(s_t, a_t) \right].$$

PPO [22] is an actor-critic method that optimizes this objective using policy gradients. It alternates between collecting trajectories using the current policy $\pi_{\theta_{\text{old}}}$ and updating the policy parameters $\theta$ as well as a value function.

In the massively parallel setting, $N$ independent copies of the environment are simulated synchronously. During the data collection phase, each of the $N$ environments executes the current policy $\pi_{\theta_{\text{old}}}$ for $K$ steps (the rollout length). This generates a batch of $N \times K$ transitions $\{(s_t, a_t, r_t, s_{t+1}, d_t)\}_{i=1..N, t=0..K-1}$, where $d_t$ indicates if state $s_{t+1}$ is terminal. This batch is then used to compute policy and value function gradients and perform updates for several epochs.

## 3.2 The Problem: Cyclical Nonstationarity with Synchronous Resets

Consider the common scenario in massively parallel RL where the PPO rollout length $K$ is chosen to be much smaller than the task horizon $H$ ($K \ll H$). This strategy aims to increase the update-to-data (UTD) ratio and overall data throughput. In standard synchronous implementations, where all $N$ parallel environments are reset only after completing their full $H$-step episode duration:

1. **Initial Synchronized Start:** All $N$ environments are initially reset, commencing their episodes at effective time $t = 0$ from an initial state distribution $\rho_0$.

2. **First Rollout Batch:** For the first PPO update, each environment executes the current policy for $K$ steps. The resulting batch of $N \times K$ transitions exclusively contains data from the time window $[0, K-1]$ of the episode.

3. **Subsequent Rollout Batches:** For the second PPO update, environments continue, and the batch is now formed from transitions within the window $[K, 2K-1]$. This pattern continues: for the $j$-th PPO update (assuming no environment has yet completed $H$ steps), the batch consists of transitions exclusively from the window $[(j-1)K, jK-1]$. Environments that terminate prematurely within a $K$-step window (e.g., due to task success/failure before $H$ steps) are typically reset to $t = 0$ and continue, slightly desynchronizing from the main $H$-step cycle but not fundamentally altering the batch-wise temporal homogeneity.

4. **Synchronized Full Reset:** After approximately $m = \lceil H/K \rceil$ rollouts, all (or most) environments will have reached or exceeded $H$ elapsed steps since their last full reset. At this point, they are all synchronously reset back to an effective time $t = 0$.

5. **Cycle Repetition:** Consequently, the PPO update following this full synchronous reset (e.g., the $(m+1)$-th update) will again process a batch exclusively composed of transitions from the $[0, K-1]$ window, mirroring the temporal origin of the very first batch.

The crucial issue stemming from this process is a cyclical nonstationarity of the training batches. While each environment eventually explores states across its entire $H$-step horizon, each individual batch used for a PPO gradient update is temporally homogeneous. It contains data only from a narrow $K$-step slice of the overall task. The learner is thus fed a data stream where the underlying state distribution within the batch shifts dramatically and predictably from one update to the next, cycling through different segments of the episode. For instance, one batch might contain only early-episode states, the next only mid-episode states, and another only late-episode states, before abruptly reverting to early-episode states after the synchronous full reset of all environments. This cyclical bias prevents the learner from accurately estimating values and advantages, likely due to catastrophic forgetting

phenomena (empirically verified in Appendix E). This leads to poor performance, instability, and a failure to learn long-horizon behaviors, effectively negating the benefits of parallelization and high UTD ratios. Empirics we collect in Sections 4 and 5 justify this claim.

## 3.3 Staggered Resets

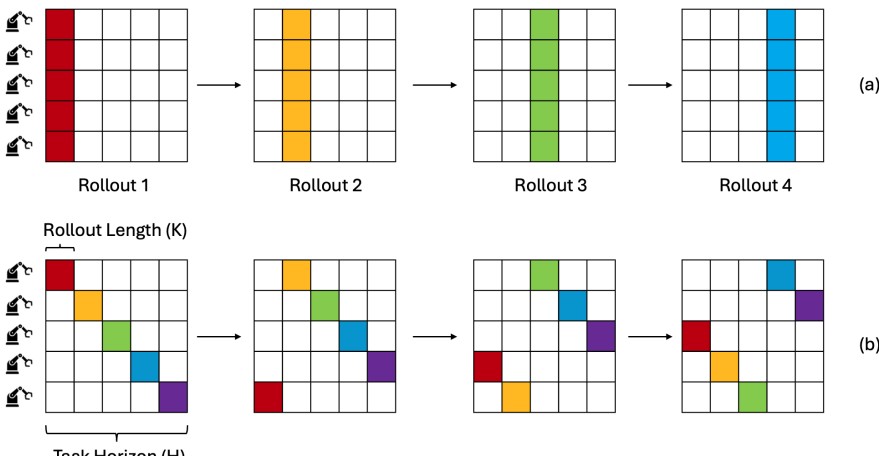

Figure 1: Data collection in massively parallel RL. Rows are environments, columns are time within task horizon $H$. Colors (red, orange, etc.) mark distinct task stages. (a) **Synchronous Resets (Naive)**: All environments start at $t = 0$ (red stage). Each rollout batch (e.g., Rollout 1: all red; Rollout 2: all orange) is temporally homogeneous. Batch content cycles through stages every $H/K$ rollouts, causing cyclical nonstationarity for the learner. (b) **Staggered Resets**: Environments start and hence at varied points in the task. Each rollout batch contains a mix of task stages (red, orange, green, blue, purple). This within-batch temporal diversity is maintained across rollouts, yielding a more stationary and representative data distribution.

To counteract this detrimental cyclical nonstationarity, we introduce **staggered resets**. The core principle is to ensure that the $N$ parallel environments are not all synchronized to the same portion of the task at the beginning of each data collection phase. Instead, they are deliberately initialized to cover diverse effective time steps within the overall task. The mechanism is straightforward:

**Initial Staggering Strategy:** Before the main training loop commences, each of the $N$ parallel environments $i$ is independently advanced (e.g., with random actions or the initial untrained policy) for a specific number of offset steps, $t_i^{\text{offset}}$. This offset $t_i^{\text{offset}}$ can be sampled uniformly from $[0, H-1]$.

However, to manage the frequency of reset and advance operations (which can be costly, especially in GPU environments if not batched), we discretize the staggering process by dividing the $N$ parallel environments into $N_B$ groups. Each environment $i$ is then assigned an offset $t_i^{\text{offset}}$ sampled from a set of discrete intervals, e.g., $\{0 \cdot S, 1 \cdot S, \ldots, (N_B - 1) \cdot S\}$, where $S$ is the stagger step size, chosen such that $N_B \cdot S \approx H$. A standard choice is $S = K$ (the PPO rollout length), leading to offsets like $\{0 \cdot K, 1 \cdot K, \ldots, \lfloor (H - K)/K \rfloor \cdot K\}$. This pre-initialization phase positions environments at distinct effective starting points within the task horizon $H$.

The choice of $N_B$ (number of distinct offset groups) balances temporal diversity against the complexity of managing reset schedules; a heuristic of $N_B \approx H/K$ often works well, ensuring coverage of the horizon within each PPO update cycle while grouping initial advance operations.

**Synchronous Rollouts with Staggered Starts:** Once initially staggered, the standard synchronous PPO data collection proceeds. All $N$ environments execute the current policy $\pi_{\theta_{\text{old}}}$ for $K$ steps. However, because each environment $i$ effectively began its current episode segment at a different offset $t_i^{\text{offset}}$ relative to the global task timeline, the aggregated batch of $N \times K$ transitions now contains experiences from a much wider and more representative range of the task horizon. For instance, if environments are staggered across multiples of $K$, the collected batch will naturally include data segments corresponding to time windows like $[0, K-1], [K, 2K-1], \ldots, [H-K, H-1]$, rather than being overwhelmingly biased towards a single segment. This concept is illustrated in Figure 1(b).

**Handling Resets During Training:**

- **End-of-Horizon Resets (End of $H$):** After an environment accumulates $H$ effective steps, it is reset to $s_0 \sim \rho_0$. These operations typically occur for entire groups of environments simultaneously, enabling efficient batching.

- **Partial Resets (Early Termination):** If an environment $j$ terminates early (e.g., success/failure before $H$ lifetime steps), it is flagged. To optimize wall-clock time by maximizing batched operations, it waits for the next scheduled "reset gate" (when a group undergoes a reset by reaching $H$ elapsed lifetime steps). At this gate, environment $j$ is reset to $s_0 \sim \rho_0$ alongside others, starting its new episode from effective time $t = 0$. This strategy minimizes the number of batched environment reset calls to align with $N_B$, thereby reducing wall-time costs. In practice, this approach effectively maintains the benefits of staggered data collection and its associated temporal diversity without significant degradation.

**Policy Updates:** The aggregated batch $\mathcal{B}$ now contains the temporally diverse transitions due to staggered starts/reset management. $\mathcal{B}$ is then used for policy and value function updates in PPO.

By ensuring temporal diversity *within each batch*, staggered resets provide the learner with a data distribution that better approximates the true state visitation distribution $\rho_\pi^{(0:H-1)}$ encountered over complete episodes. This stabilization is crucial for allowing the use of short rollouts $K$ (and thus high update-to-data ratios) without succumbing to the cyclical nonstationarity bias that plagues naive synchronous reset schemes. The improved data quality promotes more stable learning, better value estimates for states across the entire task, and ultimately, enhanced performance on long-horizon tasks. Our empirical results in Section 5 and Section 4 corroborate these benefits.

# 4 Illustrative Experiments on Toy Environments

To dissect the impact of staggered resets and pinpoint conditions where they offer maximal benefit, we conducted experiments in configurable 1-dimensional toy environments. These allow controlled variation of factors that often interact in real-world tasks. Understanding these interactions in simplified settings provides insights into the efficacy of staggered resets in more complex, high-dimensional scenarios that often exhibit similar characteristics.

## 4.1 Toy Environment Design

Our primary toy environment is a 1D chain of $B$ discrete levels. An episode spans $H$ steps, with each level covering $L = H/B$ steps. The agent's state is its current level index $b_t = \lfloor t/L \rfloor$. In each level $b$, the agent selects an action $a_t$ from a discrete set $A$. A fixed target action $a_b^* \in A$ is assigned to each level. Correct actions ($a_t = a_{b_t}^*$) yield a reward of $+0.5$, incorrect actions $-0.5$. We manipulate three key environment dynamics to explore when staggered resets are most impactful:

**Rollout Length Ratio ($K/H$):** We vary the environment task horizon $H$, exploring scenarios from $K \approx H$ (short effective skill horizon, less temporal bias) to $K \ll H$ (long effective skill horizon where cyclical nonstationarity with naive resets is hypothesized to be severe).

**Reset Dynamics:** Upon episode termination, resets can range from deterministic ($\lambda_R = 0$, always to level $b_0 = 0$) to stochastic ($\lambda_R > 0$, where $b_0 \sim \text{Poisson}(\lambda_R)$ centered around $\lambda_R$). This tests if inherent start-state randomness mitigates synchronous reset issues versus needing deliberate staggering. The term "Reset Homogeneity" in our plots (Figure 2b) refers to $2 - \lambda_R$, where higher values mean more deterministic (homogeneous) resets to $b_0 = 0$.

**Sequential Skill Gating:** Progression from level $b$ to $b + 1$ depends on a mastery threshold $k_{\text{mastery}}$ (number of correct actions $a_b^*$) and an unconditional progression probability $p_{\text{prog}}$. The agent advances if mastery is met *or* a random check ($p_{\text{prog}}$) succeeds. Varying $p_{\text{prog}} \in [0, 1]$ (where $p_{\text{prog}} = 0$ implies hard gating requiring $k_{\text{mastery}} > 0$) creates tasks from simple linear progression to those requiring sequential skill acquisition.

## 4.2 Results on Toy Environments

Experiments on these toy environments (Figure 2) characterize how different environmental factors influence the efficacy of staggered resets compared to naive synchronous rollouts, particularly highlighting the impact of data non-stationarity.

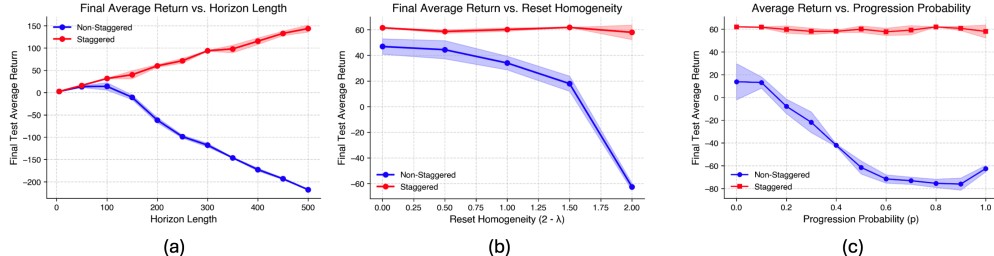

Figure 2: PPO with Non-Staggered (blue) vs. Staggered (red) resets on toy environments (mean $\pm$ 1 std dev). Staggered resets show robust performance as (a) horizon $H$ increases, (b) reset homogeneity $(2 - \lambda_R)$ increases, and (c) progression probability $(p_{\text{prog}})$ varies, unlike non-staggered PPO which degrades especially with longer horizons, more deterministic resets, and easier skill gates $(p_{\text{prog}} > 0)$.

**Horizon Length (Figure 2a):** With fixed short rollouts $(K)$, each batch of $N \times K$ transitions is temporally homogeneous, covering only a $K$-step slice of the $H$-step task. As $H$ increases, the number of distinct temporal slices in the data collection cycle $(H/K)$ also grows. This means the *periodicity* of revisiting any specific task segment (e.g., the initial states) becomes longer. The learner struggles to consolidate information and may forget what it learned about earlier segments by the time the data cycle returns to them, especially with many intermediate, distinct batch types (see Figure 12 for a visualization of this forgetting process). Staggered resets, by providing temporally diverse data within each batch, maintain high performance irrespective of $H$, effectively facilitating learning across the entire task horizon even when $K \ll H$.

**Reset Stochasticity/Homogeneity (Figure 2b):** This experiment, conducted with minimal skill gating $(p_{\text{prog}} = 1.0)$, varies the determinism of reset locations. As resets become more concentrated at the episode's start (higher "Reset Homogeneity," i.e., $\lambda_R \approx 0$), the performance of non-staggered PPO deteriorates. While some inherent randomness in reset locations $(\lambda_R > 0)$ can offer minor desynchronization and marginal performance improvements for the naive method, deliberate staggering consistently yields substantially better results. This indicates that relying on incidental environmental randomness is insufficient to overcome the core non-stationarity induced by synchronous rollouts.

**Skill Gating Dynamics (Figure 2c):** This experiment tests how $(p_{\text{prog}})$ (ease of progressing without mastery) affects data collection strategies. Staggered PPO performs robustly across all $(p_{\text{prog}})$ levels, its comprehensive sampling ensuring sufficient data for learning all skills. Non-staggered PPO's behavior is more complex: it performs best with stringent skill gates $((p_{\text{prog}} = 0))$, where the environment imposes a curriculum forcing mastery of early skills to reach later states. As $(p_{\text{prog}})$ increases (gates soften), non-staggered PPO's performance degrades. With relaxed gates, random advancement occurs, but the established data collection bias towards initial states prevents learning a coherent policy for the full task, as sporadic later-state exposure without mastery is insufficient. Consequently, the advantage of staggered resets widens as $(p_{\text{prog}})$ increases. This highlights that the cyclical non-stationarity of naive rollouts is a key limiter, especially when environmental structures (like hard gates) that aid exploration are weak.

The environmental factors varied in our toy experiments—horizon length, reset stochasticity, and skill gating—mirror complexities encountered in high-dimensional robotics tasks, such as those in ManiSkill3. For instance, long manipulation sequences often mean the task horizon $H$ is much larger than practical PPO rollout lengths $K$, exacerbating the long periodicity issue seen in Figure 2a. "Skill gates" in toy tasks are analogous to critical bottleneck sub-tasks in robotics, like achieving a stable grasp in `StackCube-v1` before attempting to lift and place. If an agent can bypass such a bottleneck (akin to high $p_{\text{prog}}$) but its experience is fed in temporally homogeneous batches, learning a robust overall strategy becomes difficult. Similarly, the degree of stochasticity in initial object poses or slight variations in robot starting conditions in ManiSkill3 tasks relates to the reset stochasticity (Figure 2b). Highly deterministic setups in robotics can make the cyclical batch problem more pronounced.

# 5   Experiments on High-Dimensional Robotics Tasks

## 5.1   Experimental Setup

Our evaluation suite includes several challenging robotics tasks from ManiSkill3 [26], a GPU-accelerated robotics framework based on SAPIEN [29]. We test on `StackCube-v1`, a manipulation task requiring an agent to stack one cube onto another; `PushT`, where a T-shaped block must be pushed to a target pose; `TwoRobotPushCube`, where two robots work together to move a cube to a goal; `Unitree Transport Box`, a humanoid task where a box must be transported to a table; and `Anymal Reach C`, where an Anymal C robot must move to a specific goal location. We also test on `MS-HumanoidWalk`, a humanoid walking control task. These environments involve high-dimensional continuous state and action spaces, providing a suitable testbed for evaluating the effectiveness of staggered resets.

## 5.2   State Visitation Dynamics in High-Dimensional Robotics

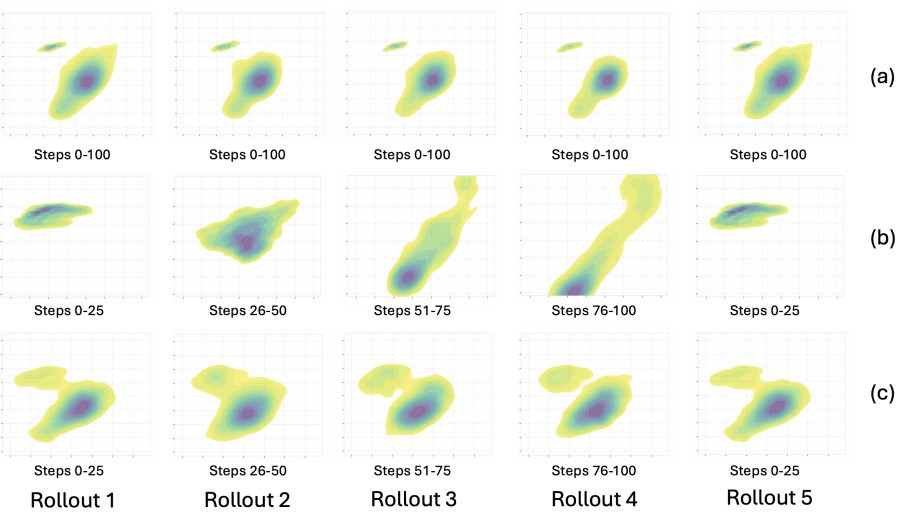

Figure 3: State visitation KDEs in `StackCube-v1` over five rollouts. (a) Long Rollout ($K = 100$): stable, broad coverage. (b) Naive Short Rollout ($K = 25$): cyclical non-stationarity, narrow/erratic coverage. (c) Staggered Short Rollout ($K = 25$): stable, diverse coverage, emulating (a) despite short trajectories.

A central premise of this work is that the common practice of using short rollouts ($K \ll H$) with naive synchronous resets in massively parallel RL leads to a problematic data generation process. This process is characterized by a temporally unstable and skewed state visitation distribution, which we hypothesize significantly impedes learning. Our proposed staggered resets intervention aims to rectify this by enabling short rollouts to yield data distributions more similar to those from longer, more informative trajectories.

To empirically assess this hypothesis, we visualize and analyze state visitation patterns on the challenging `StackCube-v1` robotics manipulation task. Figure 3 presents Kernel Density Estimates (KDEs) of visited states in a given rollout buffer, projected onto their first two principal components derived from the aggregate data of all rollouts across an entire training run. Each row corresponds to a different data collection strategy, and columns depict the evolution of the state distribution over five consecutive PPO data collection rollouts.

The Long Rollout PPO baseline ($K = 100$), shown in Figure 3a, serves as an empirical ideal, exhibiting broad and stable state coverage across all five rollouts. This pattern signifies a rich and temporally consistent data stream, achieved here at the expense of a lower UTD ratio.

Figure 3b, representing the Naive Short Rollout PPO strategy ($K = 25$), illustrates the nonstationarity in consecutive rollout buffers induced by synchronous resets. Initial rollouts show a state distribution highly concentrated near the environment's reset distribution. As training progresses, the distribu-

tion expands, but its shape and locus shift dramatically between rollouts. After the environments synchronously reset (following Rollout 4), the state distribution at Rollout 5 closely mirrors that of Rollout 1. This cyclical nonstationarity means the learner receives a constantly changing and biased view of the transition space, hindering its ability to form accurate value estimates and a robust policy.

Figure 3c depicts the state visitation dynamics for our Staggered Short Rollout PPO strategy (also $K = 25$). The distributions achieved by staggered resets qualitatively mirror the desirable breadth and consistency of the long rollout baseline (a). This empirical evidence strongly supports our central claim: staggered resets effectively counteract the cyclical non-stationarity induced by synchronous resets in short-rollout regimes.

## 5.3    Performance on High-Dimensional Robotics Tasks

To validate the effectiveness and generality of staggered resets, we conduct experiments on two distinct sets of high-dimensional robotics benchmarks, using two different state-of-the-art on-policy algorithms: Proximal Policy Optimization (PPO) and Split and Aggregate Policy Gradients (SAPG).

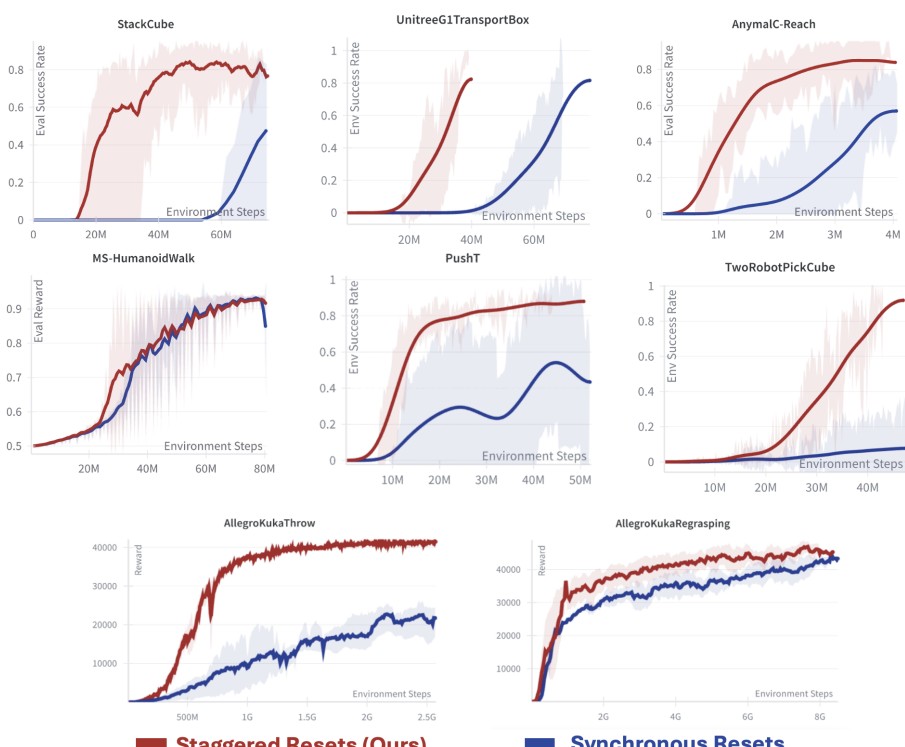

Figure 4: **Staggered Resets (Ours, red) consistently outperform Synchronous Resets (blue) across different on-policy algorithms and task suites.** Plots show the average evaluation metric (success rate or reward) vs. environment steps. Shaded areas show the standard deviation over 10 seeds. **(Top & Middle Rows)** On **PPO** with diverse ManiSkill3 tasks (`StackCube`, `PushT`, etc.), staggered resets consistently improve learning speed, final performance, and stability. Performance is comparable only on the locomotion task `MS-HumanoidWalk`, where natural desynchronization reduces the severity of the problem. **(Bottom Row)** To demonstrate algorithm-agnosticism, we evaluate on **SAPG** with challenging AllegroKuka manipulation tasks. Staggered resets again yield substantial gains in sample efficiency and final reward, confirming the generality of our approach.

### 5.3.1    Performance Improvements with PPO on ManiSkill3 Tasks

As shown in the top and middle rows of Figure 4, staggered resets achieve substantially faster convergence, higher final success rates, and greater stability on a wide range of PPO-based manipulation tasks. For instance, in `PushT` and `StackCube-v1`, staggering leads to significantly higher and

more stable final performance. Similarly, in `AnymalC-Reach` and `TwoRobotPickCube`, learning is markedly quicker and reaches a better asymptotic success rate. Crucially, with an increased evaluation of 10 seeds, the variance across runs is also substantially lower with staggered resets, confirming that our method improves overall training stability.

Interestingly, on the locomotion task `MS-HumanoidWalk`, both methods perform comparably. As discussed in our toy experiments, locomotion environments often feature shorter effective skill horizons and highly stochastic reset behaviors (e.g., the agent falling at unpredictable times). These factors induce a degree of natural desynchronization, making the cyclical batch nonstationarity less severe and thereby reducing the marginal benefit of explicit staggering. This aligns with prior work [21] that employed a form of staggering in locomotion contexts, where our analysis suggests the benefits are less pronounced compared to more complex, longer-horizon manipulation tasks.

### 5.3.2 Validating Algorithm-Agnosticism with SAPG

A central claim of our work is that the benefits of staggered resets are algorithm-agnostic, stemming from the data distribution rather than the specifics of the learning update. To substantiate this, we conduct additional experiments with a more recent on-policy algorithm, SAPG [25], on the challenging `AllegroKuka` dexterous manipulation tasks.

The results, shown in the bottom row of Figure 4, demonstrate that staggered resets significantly improve the performance of SAPG. Both `AllegroKukaThrow` and `AllegroKukaRegrasping` show dramatic improvements in sample efficiency and final asymptotic reward. Notably, these environments feature frequent early resets (e.g., on task success or object drop) and significant domain randomization, which can naturally induce some desynchronization and reduce the cyclicity problem. Despite this, staggered resets still yield a pronounced performance improvement, underscoring the robustness and general applicability of our method for improving data quality in massively parallel on-policy RL.

### 5.4 Scaling with Parallel Environments and Overcoming Performance Saturation

A key challenge in massively parallel RL is effectively utilizing increased parallelism. SAPG [25], shows that beyond a threshold, additional environments may not reduce wall-clock time and may even hurt performance due to higher gradient variance or communication overhead. We address this issue and show that staggered resets improve convergence speed across a wide range of $N$, yielding a higher marginal utility of increased environment parallelism.

Figure 5 shows the wall-clock time required to reach a fixed convergence threshold (70% success rate) as a function of the number of parallel environments $N$. Results are shown for two representative tasks: `StackCube-v1` (Figure 5a) and `Unitree G1 Transport Box` (Figure 5b).

For Naive PPO (blue curves), increasing parallel environments ($N$) initially speeds up wall-clock convergence. However, this benefit saturates or even reverses at larger $N$ (e.g., beyond $N \approx 1024$ for `StackCube-v1`, Figure 5a; degrading sharply at $N = 6144$ for `Unitree G1 Transport Box`, Figure 5b). Hence, while data throughput rises, learning efficiency diminishes due to temporally homogeneous from many synchronously reset environments, offsetting parallelism gains. Conversely, Staggered PPO (red curves) demonstrates superior scaling, with wall-clock convergence time continuing to decrease as $N$ increases across all tested values, even beyond 6000 environments.

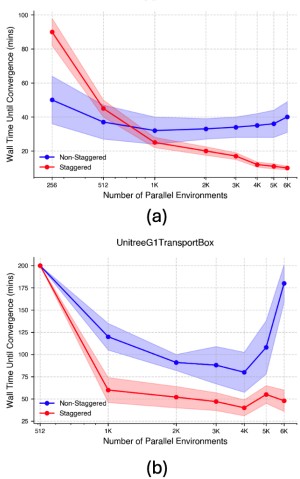

Figure 5: Wall-clock time to convergence versus number of parallel environments ($N$) for (a) `StackCube-v1` and (b) `Unitree G1 Transport Box`

## 6 Discussion

Our investigation reveals a significant challenge in modern massively parallel on-policy RL. The cyclical nonstationarity introduced by synchronous environment resets when coupled with short

rollouts ($K \ll H$) inadvertently biases the learning signal by repeatedly oversampling states from the initial segments of episodes. This bias can detrimentally affect learning stability, convergence speed, and ultimate policy quality, leading to issues like value function divergence and catastrophic forgetting, as we quantitatively demonstrate in Appendix E.

Staggered resets directly address this nonstationarity without any changes to the core learning algorithm. By deliberately initializing parallel environments at varied effective time steps within the task horizon, we ensure that each training batch encompasses a temporally diverse set of experiences. This creates a more stationary and representative data distribution for the learner. The state visitation KDEs (Figure 3) offer compelling visual evidence: staggered resets with short rollouts yield broad and stable state coverage, closely emulating the desirable properties of much longer rollouts, whereas naive short rollouts suffer from erratic, cycling state distributions.

Crucially, our work also provides empirical justification for focusing on the short-rollout ($K \ll H$) regime. As shown in our appendix experiments (Table 1), this regime is not merely a heuristic but is often optimal for wall-clock efficiency. On challenging tasks like `StackCube-v1`, short rollouts ($K = 8 - 16$) converge 2-3$\times$ faster than long rollouts while achieving the same or better final performance, reinforcing that addressing issues within this specific setting is of high practical importance.

The illustrative toy experiments (Section 4) further characterize the conditions where staggered resets are most impactful. Specifically, tasks with longer horizons relative to the rollout length, more deterministic (homogeneous) reset states, and weaker intrinsic task curricula (e.g., where agents are not strictly forced to master early skills to progress) show pronounced benefits from the explicit temporal diversification that staggering provides.

A key practical advantage of staggered resets is their ability to enhance the scalability of on-policy RL in massively parallel settings (Section 5.4, Figure 5). While naive PPO often encounters diminishing returns or even performance degradation as the number of parallel environments grows—likely due to increasingly redundant data—staggered resets facilitate continued improvements in wall-clock convergence time. This indicates a more effective utilization of parallel compute resources, as the increased data volume is also more diverse and informative.

In conclusion, staggered resets provide a robust, easily implementable, and computationally inexpensive method to significantly enhance the performance and scalability of on-policy RL. Its benefits are algorithm-agnostic, improving performance for both PPO and more recent methods like SAPG (Section 5.3.2). By directly addressing the issue of cyclical data nonstationarity, this technique allows for more stable value estimation, faster convergence, and better final policies, paving the way for more effective learning in complex, long-horizon tasks.

## 7 Acknowledgements

Stone Tao is supported in part by the NSF Graduate Research Fellowship Program grant under grant No. DGE2038238.

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

# A Motivating Experiments

## A.1 Empirical Motivation for the $K \ll H$ Regime

To motivate the massively parallel short-rollout regime, we sweep $K \in \{1, 2, 4, 8, 16, 32, 64, 100\}$ on `StackCube-v1` ($H{=}100$). Each configuration is trained for 100M steps across three seeds with identical hyperparameters. Short rollouts ($K{=}8$–16) reach the same or better final reward as long rollouts while converging 2–3$\times$ faster in wall-clock time, confirming the practical relevance of $K \ll H$ in massively parallel on-policy RL.

Table 1: Performance sweep over rollout length $K$ on `StackCube-v1`. Best results ($K{=}8$–16) in **bold**.

**(a) Final reward after 100M environment steps.**

| Metric | 1 | 2 | 4 | 8 | 16 | 32 | 64 | 100 |
|---|---|---|---|---|---|---|---|---|
| Staggered | 0.38±0.06 | 0.47±0.01 | 0.70±0.00 | **0.74±0.00** | **0.72±0.03** | 0.75±0.02 | 0.71±0.03 | 0.70±0.02 |
| Naive | 0.27±0.17 | 0.36±0.12 | 0.47±0.02 | 0.62±0.01 | **0.72±0.02** | 0.79±0.03 | 0.71±0.03 | 0.70±0.02 |

**(b) Environment steps (millions) to reach >75% success rate.**

| Metric | 1 | 2 | 4 | 8 | 16 | 32 | 64 | 100 |
|---|---|---|---|---|---|---|---|---|
| Staggered | DNC | DNC | DNC | **16.2±0.2** | 17.2±0.1 | 23.2±1.4 | 50.1±5.6 | 49.6±9.9 |
| Naive | DNC | DNC | DNC | 35.8±4.5 | **18.5±2.1** | 24.1±1.9 | 50.5±6.2 | 49.6±9.9 |

**(c) Wall-clock time (minutes) to reach >75% success rate.**

| Metric | 1 | 2 | 4 | 8 | 16 | 32 | 64 | 100 |
|---|---|---|---|---|---|---|---|---|
| Staggered | DNC | DNC | DNC | **16.6±3.2** | **15.3±1.0** | 22.5±2.3 | 40.0±8.7 | 42.6±8.7 |
| Naive | DNC | DNC | DNC | 26.2±2.1 | 31.5±13.6 | 30.0±11.9 | 41.1±4.5 | 43.2±9.9 |

**Observation.** The fastest convergence is achieved with short rollouts ($K{=}8$–16) on an NVIDIA RTX-4090 GPU, matching Rudin et al. (2021) and Singla et al. (2024). Training with $K{=}1$–4 fails to converge (DNC). Final reward saturates beyond $K{=}8$, reinforcing that modern massively parallel on-policy systems naturally operate in the $K \ll H$ regime.

# B Implementation Details

## B.1 Staggered Reset Implementation Details

The staggered reset mechanism aims to distribute the effective starting timesteps of the $N$ parallel environments across the task horizon $H$. This was achieved by dividing environments into $N_B = \lceil H/K \rceil$ groups, with each group $j$ starting its first "effective" episode step after an initial offset of $j \cdot K$ simulation steps (typically performed with random actions or the initial policy). This ensures that each PPO batch contains data from various segments of the task horizon.

## B.2 Details on Toy Environments

The toy environments described in Section 4 were designed to isolate and study the effects of data nonstationarity under different environmental conditions. See Table 2 for the hyperparameters chosen for PPO in the toy environment. We describe more concretely the ablation and environment implementation details below.

### B.2.1 Environment Dynamics

The environment is a 1-dimensional chain of $B$ discrete levels or "blocks". An episode lasts for a maximum of $H$ time steps. Each level $b \in \{0, \ldots, B-1\}$ covers $L = H/B$ steps. The agent's state $s_t$ is its current level index $b_t = \lfloor \text{elapsed\_steps}_t / L \rfloor$. At each time step $t$, the agent, being in

level $b_t$, chooses an action $a_t$ from a discrete set of $A_c$ categories (e.g., $A_c = 20$). Each level $b$ has a pre-assigned target action $a_b^* \in \{0, \ldots, A_c - 1\}$. The reward function is:

$$r(s_t, a_t) = \begin{cases} +0.5 & \text{if } a_t = a_{b_t}^* \\ -0.5 & \text{if } a_t \neq a_{b_t}^* \end{cases}$$

The episode terminates if $\text{elapsed\_steps}_t \geq H$.

### B.2.2 Further Details on Ablations on Toy Environments

The following parameters were varied to create the different experimental conditions shown in Figure 2:

- **Horizon Length ($H$ vs. $K$):** (Figure 2a) The task horizon $H$ (max_steps in code) was varied across values [50, 100, 200, 300, 400, 500]. The PPO rollout length $K$ (num_steps in PPO loop, i.e., buffer size per environment before update) was kept fixed at $K = 5$. The block length $L$ was also fixed at 5. For this experiment, skill gating was moderate ($p_{\text{prog}} = 0.5$, $k_{\text{mastery}} = 3$) and reset was deterministic ($\lambda_R = 0$).

- **Reset Stochasticity/Homogeneity ($\lambda_R$):** (Figure 2b) Upon episode termination, the reset mechanism was varied. The parameter $\lambda_R$ (reset_stochasticity_lambda in code) controls the mean of a Poisson distribution from which the starting block $b_0$ is sampled, i.e., $b_0 \sim \text{Poisson}(\lambda_R)$, clamped to $[0, B - 1]$. $\lambda_R = 0$ corresponds to a deterministic reset to $b_0 = 0$. The "Reset Homogeneity" axis in the plot is $2.0 - \lambda_R$ for visualization purposes (higher values = more deterministic starts at $b_0 = 0$). $\lambda_R$ was varied in $[0.0, 0.1, \ldots, 1.0]$. For this experiment, $H = 50$, $L = 5$, $K = 5$, $p_{\text{prog}} = 1.0$ (easy progression), $k_{\text{mastery}} = 3$.

- **Skill Gating Dynamics ($p_{\text{prog}}$):** (Figure 2c) Progression from the current block $b$ to $b + 1$ (when enough steps within block $b$ have nominally passed to enter $b + 1$) occurs if either:
  1. The agent has achieved "mastery" in block $b$, defined as making at least $k_{\text{mastery}}$ correct actions $a_b^*$ within block $b$ during the current episode. ($k_{\text{mastery}} = 3$ was used).
  2. A random chance $p_{\text{prog}}$ for unconditional progression succeeds.

  The probability $p_{\text{prog}}$ (progression_prob in code) was varied in $[0.0, 0.1, \ldots, 1.0]$. $p_{\text{prog}} = 0$ means hard gating requiring mastery. For this experiment, $H = 200$, $L = 5$, $K = 5$, $\lambda_R = 0$.

### B.3 Implementation Details and Hyperparameters for ManiSkill Experiments

This section details the Proximal Policy Optimization (PPO) configuration for experiments on ManiSkill robotics tasks (Section 5), utilizing ManiSkill3 [26] for GPU-accelerated simulation. The exact PPO implementation is based on the one provided by ManiSkill3 baselines, which is based on LeanRL and CleanRL. Table 3 provides a comprehensive list of hyperparameters.

Table 2: PPO and Environment Hyperparameters for Toy Environment Experiments. Shaded rows categorize parameters. Values listed are defaults; specific sweeps varied $H$, $p_{\mathrm{prog}}$, or $\lambda_R$ as detailed in text and figures.

| Hyperparameter | Value |
|---|---|
| **PPO Algorithm Core Settings** | |
| Learning Rate | $3 \times 10^{-4}$ |
| Discount Factor ($\gamma$) | 0.99 |
| GAE Lambda ($\lambda$) | 0.95 |
| PPO Rollout Length ($K$) | 5 steps per environment |
| Number of Parallel Environments ($N$) | 512 |
| Total Training Updates | 150 |
| Update Epochs | 4 |
| Number of Minibatches | 4 (Minibatch size: $(512 \times 5)/4 = 640$) |
| PPO Clipping Coefficient ($\epsilon$) | 0.2 |
| Value Function Loss Coefficient | 0.5 |
| Entropy Bonus Coefficient | 0.01 |
| Max Gradient Norm | 0.5 |
| **Network Architecture (Actor & Critic MLP)** | |
| Input | Current block index (integer state) |
| Embedding Layer | Input block index to 64-dim embedding |
| Hidden Layers | 4 |
| Units per Hidden Layer | 256 |
| Activation Function | ReLU |
| Policy Output | Categorical distribution over actions |
| Value Output | Scalar state value |
| **Optimization** | |
| Optimizer | Adam |
| **Toy Environment Base Parameters (Defaults for Sweeps)** | |
| Episode Horizon ($H$) | Varied (e.g., 50, 100, 200, 375 for specific experiments) |
| Block Length ($L$) | 5 steps |
| Number of Action Categories ($A_c$) | 20 |
| Reward for Correct Action | +0.5 |
| Reward for Incorrect Action | -0.5 |
| Success Definition | Agent is in the final block at episode end |
| Skill Gating: Progression Prob. ($p_{\mathrm{prog}}$) | Varied (0.0 to 1.0) |
| Skill Gating: Mastery Threshold ($k_{\mathrm{mastery}}$) | 3 correct actions |
| Reset Stochasticity ($\lambda_R$) | Varied (Poisson mean for start block, 0.0 for deterministic) |
| **Staggered Resets Mechanism (When Enabled)** | |
| Number of Stagger Blocks ($N_B$) | $\lceil H/K \rceil = \lceil \text{Episode Horizon}/5 \rceil$ |
| Stagger Step Size ($S$) | $K = 5$ |

Table 3: PPO Hyperparameters for ManiSkill Experiments. Common settings are listed first, followed by per-environment variations where applicable. Shaded rows categorize parameters.

| Hyperparameter | Value / Per-Environment Specification |
|---|---|
| **PPO Algorithm Core Settings** | |
| Learning Rate | $3 \times 10^{-4}$ |
| Update Epochs | 4 |
| Number of Minibatches | 32 |
| PPO Clipping Coefficient ($\epsilon$) | 0.2 |
| Value Function Loss Coefficient | 0.5 |
| Entropy Bonus Coefficient | 0.005 |
| Max Gradient Norm | 0.5 |
| Advantage Normalization | True (Per minibatch) |
| Target KL for Early Stopping | 0.1 |
| **Network Architecture (Actor & Critic MLP)** | |
| Hidden Layers | 3 |
| Units per Hidden Layer | [256, 256, 256] |
| Activation Function | Tanh |
| Weight Initialization | Orthogonal |
| Policy Output | Gaussian mean, learnable state-independent log std. dev. |
| **Optimization** | |
| Optimizer | Adam |
| Adam Epsilon | $1 \times 10^{-5}$ |
| Learning Rate Annealing | False |
| **Environment Interaction & Data Collection (Common)** | |
| Total Training Timesteps | $2 \times 10^8$ |
| Partial Resets (Training) | True |
| Evaluation Environments | 128 |
| Evaluation Partial Resets | False |
| Observation Normalization | Via environment wrappers / running mean & std |
| Reward Scaling | Environment-dependent (aim for std approx. 1) |
| **Staggered Resets Mechanism (When Enabled)** | |
| Staggering Mode | Uniform distribution of start times |
| Number of Stagger Blocks ($N_B$) | $\lceil H/K \rceil$ (Task Horizon / Rollout Length) |
| Stagger Step Size ($S$) | $K$ (Rollout Length) |
| **Per-Environment Specific Hyperparameters** | |
| **Parameter** | **Values for: StackCube / PushT / AnymalC / HumanoidWalk / TwoRobotCube / UnitreeBox** |
| Rollout Length ($K$) | 8 / 8 / 16 / 64 / 16 / 32 |
| Task Horizon ($H$) | 100 / 100 / 200 / 1000 / 100 / 500 |
| Discount Factor ($\gamma$) | 0.8 / 0.99 / 0.99 / 0.97 / 0.8 / 0.8 |
| GAE Lambda ($\lambda$) | 0.9 / 0.9 / 0.95 / 0.9 / 0.9 / 0.9 |
| Num. Parallel Env. ($N$) | 4096 / 4096 / 512 / 4096 / 2048 / 1024 |

## B.4 Implementation Details and Hyperparameters for SAPG Experiments

This section details the configuration for the experiments on the AllegroKuka dexterous manipulation tasks (Section 5.3), which use the Split and Aggregate Policy Gradients (SAPG) algorithm. The experiments were conducted using the IsaacGym simulator. To ensure a fair comparison and isolate the effect of staggered resets, we adopt the official hyperparameters reported in the original SAPG paper [25]. Table 4 provides a comprehensive list of these hyperparameters. Our method introduces only the staggered reset mechanism on top of this baseline.

Table 4: SAPG Hyperparameters for AllegroKuka Experiments, based on [25]. Shaded rows categorize parameters.

| Hyperparameter | Value |
|---|---|
| **SAPG Algorithm Core Settings** | |
| Learning Rate | $1 \times 10^{-4}$ |
| Update Epochs | 2 |
| Mini-batch Size | Num. Parallel Env. $\times$ 4 |
| Clipping Coefficient ($\epsilon$) | 0.1 |
| Value Function Loss Coefficient | 4.0 |
| Entropy Bonus Coefficient | 0 (default, see [25] for tuning) |
| Max Gradient Norm | 1.0 |
| KL Threshold for LR Update | 0.016 |
| Bounds Loss Coefficient | 0.0001 |
| **Network Architecture (Recurrent Actor & Critic)** | |
| Observation Preprocessor | MLP with hidden layers [768, 512, 256] |
| Policy/Value Core | 1-layer LSTM with 768 hidden units |
| Activation Function | ELU |
| Policy Output | Gaussian mean, learnable state-independent std. dev. |
| **Optimization** | |
| Optimizer | Adam |
| **Environment Interaction & Data Collection** | |
| Number of Parallel Env. ($N$) | 24576 |
| Rollout Length ($K$) | 16 |
| Discount Factor ($\gamma$) | 0.99 |
| GAE Lambda ($\lambda$) | 0.95 |
| LSTM Sequence Length | 16 |
| **Staggered Resets Mechanism (When Enabled)** | |
| Staggering Mode | Uniform distribution of start times |
| Number of Stagger Blocks ($N_B$) | $\lceil H/K \rceil$ (Task Horizon / Rollout Length) |
| Stagger Step Size ($S$) | $K$ (Rollout Length) |

# C  Additional Results on Toy Environments

To further illustrate the impact of synchronous versus staggered resets on the data distribution and learning progress, we visualize the training dynamics in one of the toy environments (specifically, $H = 200, L = 5, K = 5, p_{\text{prog}} = 0.5, \lambda_R = 0$). Figure 6 shows the training average accuracy and the mean distribution of environments across different blocks (states) over the course of training updates. For these experiments, we define accuracy as the rolling percentage of correct one-hot action guesses from the PPO agent.

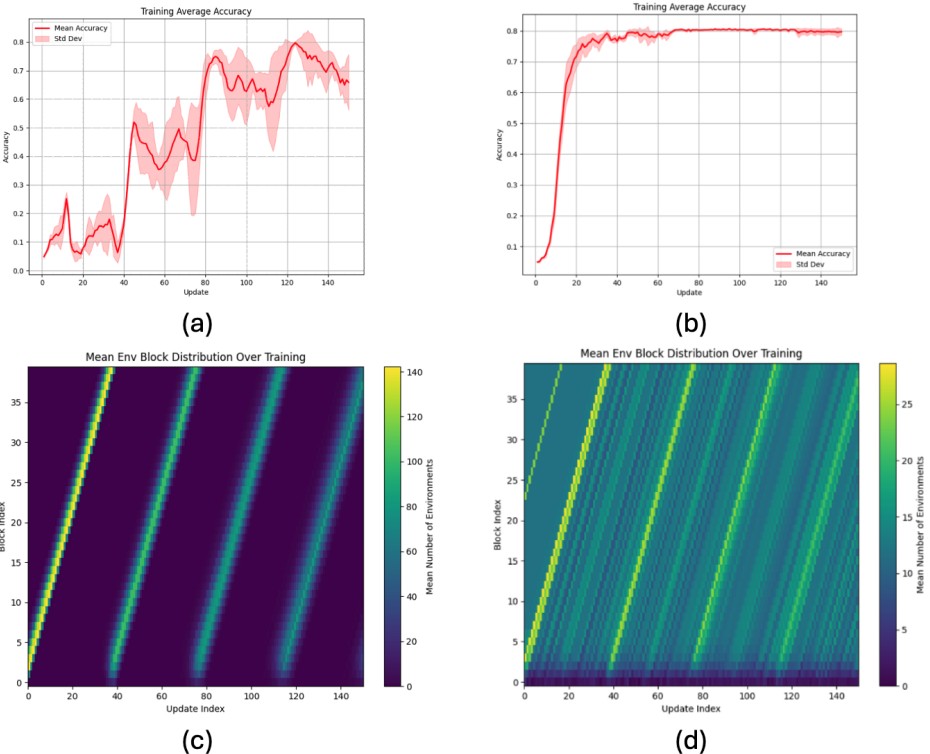

Figure 6: Comparison of training dynamics in a toy environment with (a, c) naive synchronous resets versus (b, d) staggered resets. **(a) & (b):** Training average accuracy over 150 PPO updates. With naive resets (a), accuracy is unstable and struggles to converge. With staggered resets (b), accuracy rises quickly and stabilizes at a high level. **(c) & (d):** Heatmaps showing the mean number of environments occupying each block (y-axis) at each PPO update index (x-axis). (c) With naive synchronous resets, environments progress through blocks in tight, synchronized waves. After approximately 40 updates (when $H/K = 200/5 = 40$ rollouts complete an episode), all environments abruptly reset to block 0, leading to a cyclical pattern where training batches are temporally homogeneous (all early-episode, then all mid-episode, etc.). (d) With staggered resets, the distribution of environments across blocks is far more uniform at any given update index. This indicates that each training batch contains a mix of experiences from different stages of the episode, leading to a more stationary data distribution for the learner.

Subplots (c) and (d) in Figure 6 are heatmaps where the x-axis represents the PPO training update index, the y-axis represents the block index (state) within the toy environment's episode, and the color intensity indicates the mean number of parallel environments present in that block at that specific training update.

**Naive Synchronous Resets (Figure 6c)**    The heatmap for naive synchronous resets clearly shows distinct diagonal bands. Each band signifies that the cohort of parallel environments is synchronously progressing through the episode's blocks. A crucial observation is the abrupt termination of these bands followed by an immediate restart from block 0 (the bottom of the y-axis). This occurs approximately every 40 updates, corresponding to the episode horizon ($H = 200$) divided by the

rollout length ($K = 5$). This pattern visually confirms the cyclical nonstationarity discussed in Section 3.2: at any given update, the training batch is predominantly composed of states from a narrow segment of the episode, and this segment predictably cycles. For instance, for updates 1-5, data is from blocks near 0-4; for updates 35-40, data is from blocks near 35-39; then at update 41, data abruptly shifts back to blocks 0-4.

**Staggered Resets (Figure 6d)** In contrast, the heatmap for staggered resets shows a much more diffuse and uniform pattern. While environments still progress through blocks (indicated by the general upward-right trend), there is no global, abrupt reset of all environments. At any given PPO update index, environments are distributed across a wide range of blocks. This means that each training batch collected under staggered resets contains a temporally diverse mix of experiences— some from early parts of episodes, some from middle, and some from later parts. This significantly reduces the cyclical nonstationarity of the data fed to the PPO algorithm.

**Impact on Learning (Figure 6a and 6b)** The consequences of these different state visitation dynamics are evident in the training accuracy plots. With naive synchronous resets (Figure 6a), the learning curve for average accuracy is highly erratic, exhibiting periodic dips and slow overall improvement, struggling to consistently achieve high accuracy. This instability likely results from the PPO learner trying to adapt to the rapidly shifting data distributions. Conversely, with staggered resets (Figure 6b), the average accuracy rises smoothly and rapidly, quickly converging to a high and stable level. This demonstrates that providing the learner with more temporally diverse and stationary batches facilitates more effective and stable learning.

### C.1 State Visitation Distribution Analysis

We provide further detail on the state visitation dynamics in our toy environments by visualizing the mean environment block distribution over training updates (similar to Figure 6c and 6d) while systematically varying key environmental parameters. For all visualizations in this section, the PPO rollout length $K = 5$, number of parallel environments $N = 512$, total training updates shown are 150, environment block length $L = 5$, number of action categories $A_c = 20$, and mastery threshold $k_{\text{mastery}} = 3$, unless specified otherwise. Base PPO hyperparameters are detailed in Appendix B.2. Each figure's top row shows results for naive synchronous resets, and the bottom row shows results for staggered resets.

### C.2 Impact of Progression Probability ($p_{\text{prog}}$)

Figure 7 illustrates how the probability of unconditional progression, $p_{\text{prog}}$, affects state visitation. The experiment uses a fixed horizon $H = 200$ and deterministic resets ($\lambda_R = 0$).

**Non-Staggered Resets (Top Row):**

- $p_{\text{prog}} = 0.0$ **(Left):** With hard skill gating, environments get stuck at early blocks if they fail to achieve mastery. The heatmap shows most environments concentrated at very low block indices, with only very few managing to progress. The cyclical reset pattern is still evident for those that do run the full course or get reset.

- $p_{\text{prog}} = 0.5$ **(Center):** Partial random progression allows more environments to reach later blocks, but the density remains higher at earlier stages due to the gating. The cyclical nature of resets is clear.

- $p_{\text{prog}} = 1.0$ **(Right):** Environments progress freely through blocks irrespective of mastery. This results in the most pronounced cyclical bands, as all environments synchronously march through the episode blocks and reset together.

**Staggered Resets (Bottom Row):** Across all values of $p_{\text{prog}}$, staggered resets maintain a significantly more uniform distribution of environments across different blocks at any given training update. While a lower $p_{\text{prog}}$ means individual environments might take longer or struggle more to traverse all blocks within their own episode lifetime, the staggering ensures that the batch fed to the learner still contains diverse experiences. The overall "texture" of the heatmap becomes denser as $p_{\text{prog}}$ increases, indicating that more environments are successfully exploring the full range of blocks over time, but the crucial within-batch temporal diversity is preserved by the staggered mechanism itself.

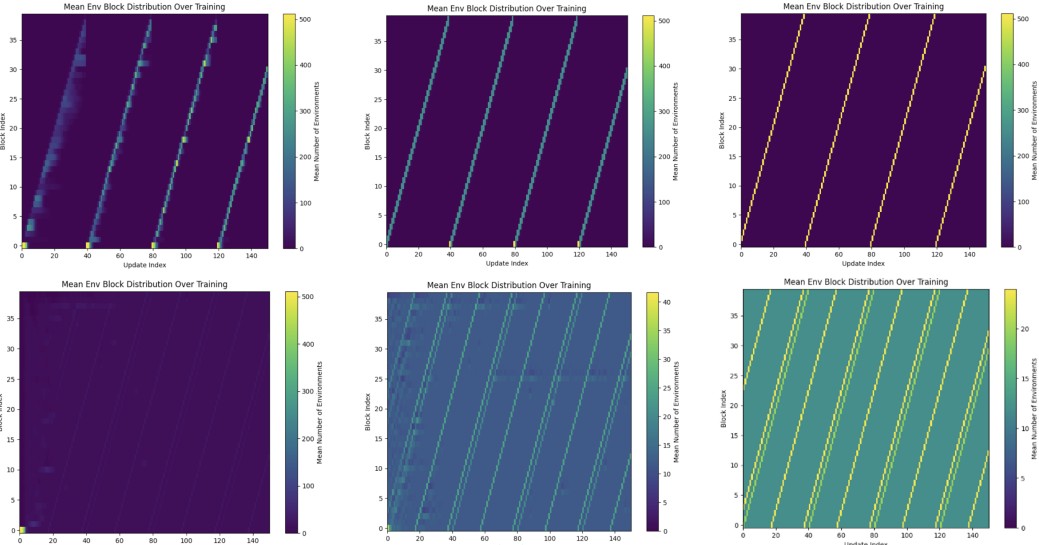

Figure 7: Mean environment block distribution over training for varying progression probabilities ($p_{\text{prog}}$). **Top Row (Non-Staggered):** From left to right, $p_{\text{prog}} = 0.0, 0.5, 1.0$. **Bottom Row (Staggered):** From left to right, $p_{\text{prog}} = 0.0, 0.5, 1.0$. With non-staggered resets, low $p_{\text{prog}}$ (hard skill gating) leads to environments bunching at early blocks, with sparse exploration of later stages. As $p_{\text{prog}}$ increases, environments progress more freely, but the strong cyclical reset pattern remains. With staggered resets, coverage is more uniform across updates regardless of $p_{\text{prog}}$, though higher $p_{\text{prog}}$ allows environments to explore the full range of blocks more rapidly within their individual (staggered) episode timelines.

### C.3  Impact of Episode Horizon Length ($H$)

Figure 8 shows the effect of varying the episode horizon length $H$. For these plots, progression probability $p_{\text{prog}} = 0.5$ and reset stochasticity $\lambda_R = 0$ are fixed. The number of blocks $B = H/L$ changes with $H$.

**Non-Staggered Resets (Top Row):**

- $H = 10$ (**Left**): With a very short horizon, the full episode cycle $H/K = 10/5 = 2$ updates. The cyclical pattern appears as very rapid, almost vertical bands. While cyclical, all parts of this very short episode are revisited extremely frequently. The y-axis shows only 2 blocks ($10/5$).

- $H = 150$ (**Center**): The cycle period is $150/5 = 30$ updates. Clear diagonal bands show synchronous progression and reset. The y-axis spans 30 blocks.

- $H = 375$ (**Right**): The cycle period becomes $375/5 = 75$ updates. The bands are elongated, indicating a longer time between revisiting the same episode phase. The y-axis spans 75 blocks. This long periodicity is hypothesized to be particularly detrimental.

**Staggered Resets (Bottom Row):**    Staggered resets consistently provide diverse state visitations within each batch, regardless of the horizon length $H$. The heatmaps show that environments are distributed across the available blocks (which scale with $H$) at each update. This ensures the learner receives a more stationary data stream, which is particularly beneficial for longer horizons where the non-staggered approach suffers from infrequent revisitation of early-episode states.

### C.4  Impact of Reset Stochasticity ($\lambda_R$)

Figure 9 visualizes the influence of reset stochasticity, $\lambda_R$, which controls the mean of a Poisson distribution for sampling the starting block upon reset. Here, $H = 200$ and $p_{\text{prog}} = 0.5$.

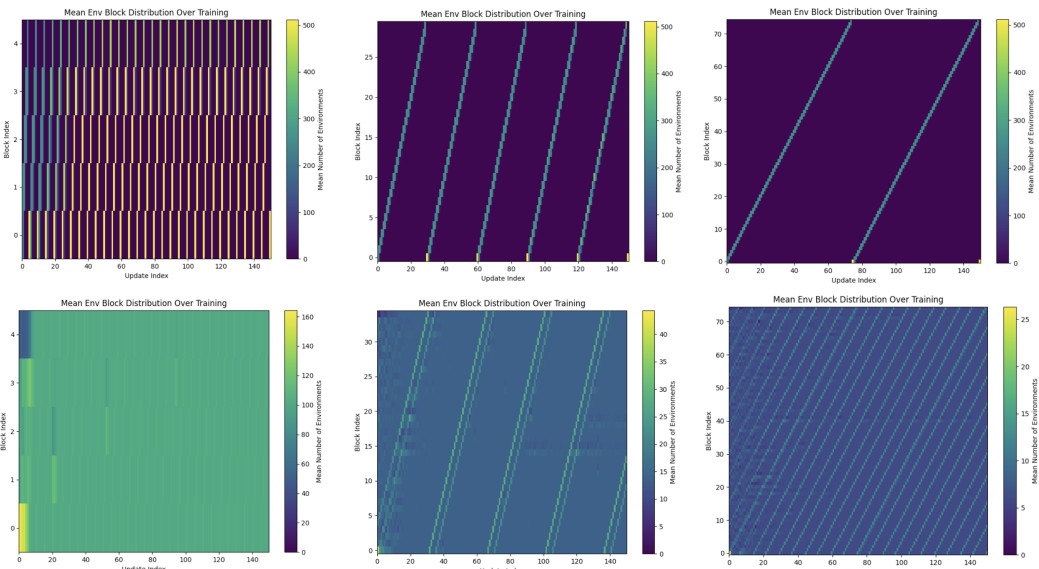

Figure 8: Mean environment block distribution over training for varying episode horizon lengths ($H$). **Top Row (Non-Staggered):** From left to right, $H = 10, 150, 375$. (Note: y-axis Block Index scales with $H$). **Bottom Row (Staggered):** From left to right, $H = 10, 150, 375$. For non-staggered resets, shorter horizons ($H = 10$) lead to very rapid cycles ($H/K = 10/5 = 2$ updates per cycle). As $H$ increases, the period of these cycles becomes longer, potentially exacerbating learning instability. Staggered resets maintain uniform coverage irrespective of $H$.

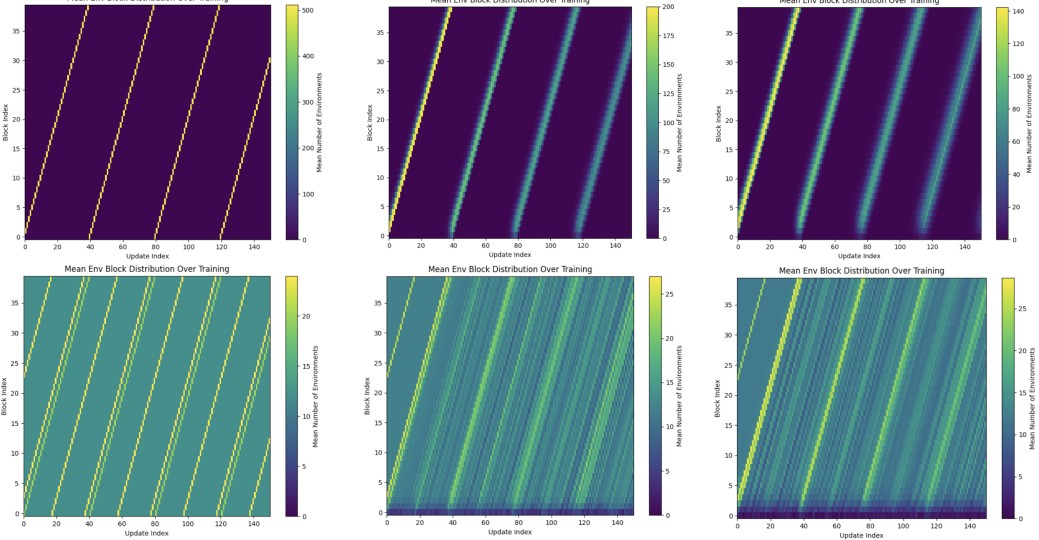

Figure 9: Mean environment block distribution over training for varying reset stochasticity ($\lambda_R$). **Top Row (Non-Staggered):** From left to right, $\lambda_R = 0.0, 1.0, 2.0$. **Bottom Row (Staggered):** From left to right, $\lambda_R = 0.0, 1.0, 2.0$. For non-staggered resets, increasing $\lambda_R$ slightly "fuzzes" the start of each cycle after a mass reset, but the overall cyclical progression remains. Staggered resets maintain uniform coverage; $\lambda_R$ primarily influences the initial state distribution within each environment's individual staggered timeline.

**Non-Staggered Resets (Top Row):**

- $\lambda_R = 0.0$ **(Left):** Deterministic reset to block 0. This results in the sharpest cyclical bands, as all environments restart precisely from the same initial state after completing an episode.

- $\lambda_R = 1.0$ **(Center) and** $\lambda_R = 2.0$ **(Right):** Stochastic resets mean environments restart from a distribution of initial blocks centered around block 0 (due to Poisson sampling with mean $\lambda_R$, then clamped). This causes the beginning of each major cycle (after most environments have run for $H$ steps) to appear slightly "fuzzier" or more spread out near block 0. However, once this initial phase passes, the environments that didn't terminate early tend to re-synchronize in their progression, and the cyclical bands through the bulk of the episode remain prominent. The inherent reset stochasticity offers only a minor and temporary desynchronization.

**Staggered Resets (Bottom Row):**   With staggered resets, the distribution of environments across blocks remains largely uniform over updates, irrespective of $\lambda_R$. The primary mechanism for achieving temporal diversity in batches is the explicit staggering of episode start times. The reset stochasticity parameter $\lambda_R$ further diversifies the exact starting block for an environment when its individual (staggered) episode concludes and it resets, but it does not fundamentally change the broad, uniform coverage ensured by the staggering mechanism itself. Staggered resets are effective even with deterministic environment resets ($\lambda_R = 0.0$).

These visualizations across different parameter sweeps consistently highlight the ability of staggered resets to create more temporally diverse and stationary training batches compared to naive synchronous resets, providing a more stable learning signal for the on-policy RL agent.

# D   Wall-Time Results and Additional Analysis

## D.1   Wall-Time Analysis for Staggering Granularity ($N_B$)

A critical aspect of staggered resets is balancing the desired temporal diversity of training batches with the computational overhead associated with environment reset operations. While an ideal scenario might involve resetting each environment at a unique timestep (effectively $N_B \approx N$, or even finer if $N_B \approx H/\text{sim\_dt}$), frequent, unbatched 'env.reset()' calls can be costly, especially in GPU-accelerated simulations. The parameter $N_B$, representing the number of distinct stagger groups or "blocks," controls this trade-off. A smaller $N_B$ means fewer, larger groups of environments are reset/advanced synchronously, reducing reset call frequency but potentially coarsening the approximation of a truly staggered (temporally diverse) data distribution.

We empirically investigated this trade-off on the `StackCube-v1` task ($H = 100$, with PPO rollout $K = 8$, thus $H/K \approx 12.5$) by measuring the wall-clock time to reach a 70% success rate while varying $N_B$. Figure 10 illustrates the results.

The findings, shown in Figure 10, indicate:

- **Few Blocks ($N_B \approx 1$):** When $N_B = 1$, all environments are effectively synchronized, resembling the naive reset scheme. This results in the slowest wall-clock convergence due to the detrimental effects of cyclical batch nonstationarity.

- **Moderate Blocks ($N_B \approx H/K$):** As $N_B$ increases, wall-clock time to convergence rapidly decreases. The optimal performance is typically observed when $N_B$ is in the vicinity of $H/K$. For `StackCube-v1` with $K = 8$, this optimal is around $N_B \approx 10 - 13$. This granularity provides sufficient temporal diversity in training batches to stabilize learning and accelerate convergence.

- **Many Blocks ($N_B \gg H/K$):** Further increasing $N_B$ beyond $H/K$ leads to marginal improvements or even a slight degradation in wall-clock convergence time. While providing finer-grained staggering, the overhead of managing and executing resets for many small, distinct groups may start to outweigh the benefits from any additional (and likely minimal) gains in data diversity.

This analysis demonstrates that a judicious choice of $N_B$, typically around $H/K$, allows us to effectively approximate the benefits of a continuously staggered reset distribution (where each environment could theoretically start at any unique step within $H$) while maintaining wall-clock efficiency. This approach strikes a practical balance, achieving most of the sample efficiency gains from temporal diversity without incurring the potentially significant computational costs of excessively

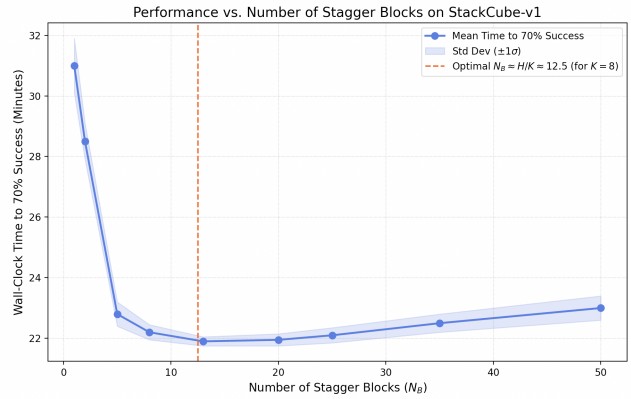

Figure 10: Wall-clock time to convergence (70% success on `StackCube-v1`) as a function of the number of stagger blocks ($N_B$). Performance, measured by faster convergence (lower wall-clock time), improves significantly as $N_B$ increases from 1 (naive synchronous resets) towards $N_B \approx H/K$ (here, $H = 100, K = 8$, so $H/K \approx 12.5$). Beyond this point, further increasing $N_B$ yields diminishing returns or even a slight increase in wall-clock time, likely due to the overhead of managing more numerous, smaller reset groups outweighing marginal gains in temporal diversity. This demonstrates that a moderate number of stagger blocks (e.g., $N_B \approx H/K$) effectively balances temporal diversity benefits with wall-time efficiency, approximating a continuously staggered reset distribution without incurring prohibitive reset costs.

frequent or unbatched reset operations. The default setting in our experiments for staggering (see Appendix B.3) is chosen based on this principle, typically defaulting to $\lfloor H/K \rfloor$.

## D.2  Wall-Clock Training Time for ManiSkill Environments

Beyond improvements in sample efficiency (i.e., performance per environment step), staggered resets also demonstrate significant advantages in terms of wall-clock training time. Figure 11 presents a comparison of evaluation success rates against wall-clock time for several challenging ManiSkill tasks.

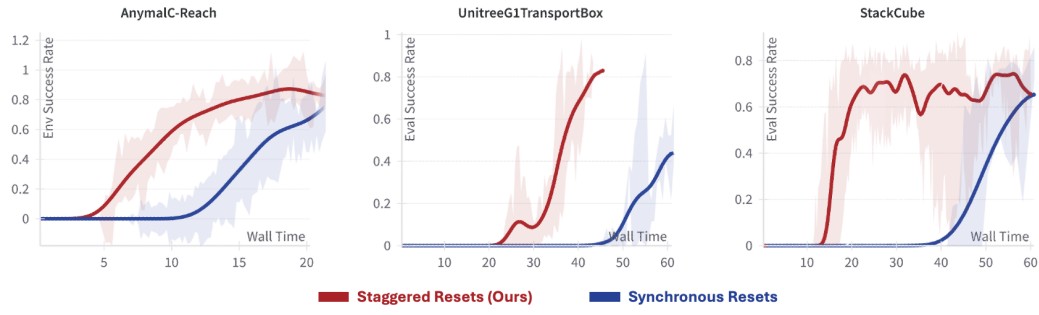

Figure 11: Comparison of evaluation success rates versus wall-clock training time for Staggered Resets (Ours, red) and Synchronous Resets (blue) on three ManiSkill tasks: `AnymalC-Reach`, `UnitreeG1TransportBox`, and `StackCube`. The x-axis represents wall-clock time (units may vary per plot, e.g., minutes or hours, but relative comparison is key). Staggered resets consistently achieve higher success rates faster, or reach comparable success rates in significantly less wall-clock time than synchronous resets. This highlights that the benefits of improved data quality and learning stability from staggered resets translate directly into more efficient use of compute time.

As illustrated in Figure 11, policies trained with staggered resets (red curves) consistently achieve target success rates in substantially less wall-clock time compared to those trained with naive

synchronous resets (blue curves). For instance, in `AnymalC-Reach`, staggered resets reach over 80% success much earlier than synchronous resets begin to show significant learning. Similarly, for `UnitreeG1TransportBox` and `StackCube`, the learning curves for staggered resets are considerably steeper when plotted against wall time, indicating faster convergence to high-performing policies.

This empirical evidence supports the conclusion that the improved sample efficiency and learning stability afforded by staggered resets directly translate into reduced overall training time, making the technique not only more data-efficient but also more computationally efficient in practice for achieving desired performance levels on complex robotics tasks. The overhead of managing staggered resets is outweighed by the gains from more effective learning per unit of time.

# E Quantitative Analysis of Cyclical Nonstationarity Effects

In the main text (Sections 1, 3.2, and 4.2), we claim that the cyclical nonstationarity induced by synchronous resets can destabilize value function learning, induce policy oscillations, and cause catastrophic forgetting. This appendix provides direct quantitative evidence for these claims using the toy environment described in Section 4.1. We compare two settings: one using our proposed staggered resets and one using naive synchronous resets, with a moderate skill gate (`progression_prob=0.1`) to highlight the differences.

### E.0.1 Metric Computation Details

To substantiate our claims, we introduce three quantitative metrics computed during the toy environment training runs. The following descriptions detail their implementation.

**Value Function Estimation Error.** To measure the stability of the critic network, we track the value function loss at each PPO update. This is computed as the Mean Squared Error (MSE) between the critic's value predictions for the states in the rollout buffer, $V(s_t)$, and the empirical Monte-Carlo returns calculated via Generalized Advantage Estimation (GAE). Large spikes in this value indicate that the critic is struggling to predict returns, often due to a sudden shift in the underlying data distribution.

**Policy Oscillations.** To quantify the magnitude of policy changes between updates, we measure the approximate KL divergence between the policy before an update ($\pi_{\text{old}}$) and the policy after the update ($\pi_{\text{new}}$). This is calculated as the mean squared difference between the log-probabilities of the actions taken during the rollout: $0.5 \cdot \mathbb{E}[(\log \pi_{\text{new}}(a|s) - \log \pi_{\text{old}}(a|s))^2]$. Consistently high or spiky KL values suggest the agent is making large, unstable updates rather than smooth, incremental improvements.

**Catastrophic Forgetting.** To directly measure the agent's ability to retain knowledge of early-episode skills while training on late-episode data, we construct a "forgetting matrix." The process is as follows:

1. At every training update $t$, we record the agent's average accuracy on each discrete block $b$ of the task, forming an accuracy matrix $A(t, b)$.

2. For each block, we compute the best-so-far accuracy achieved up to the current update: $A_{\text{best}}(t, b) = \max_{t' \leq t} A(t', b)$.

3. We define forgetting at update $t$ for block $b$ as the difference between the peak performance and the current performance: $\text{Forgetting}(t, b) = A_{\text{best}}(t, b) - A(t, b)$.

A high value in this matrix (visualized as a bright color in the plots) signifies that the agent previously had high accuracy on a specific block but has since "forgotten" how to solve it, resulting in a performance drop. A dark matrix indicates stable knowledge retention.

### E.0.2 Analysis of Results

Figures 12 and 13 provide compelling evidence for the detrimental effects of cyclical nonstationarity. The analysis below directly compares the corresponding plots from each figure.

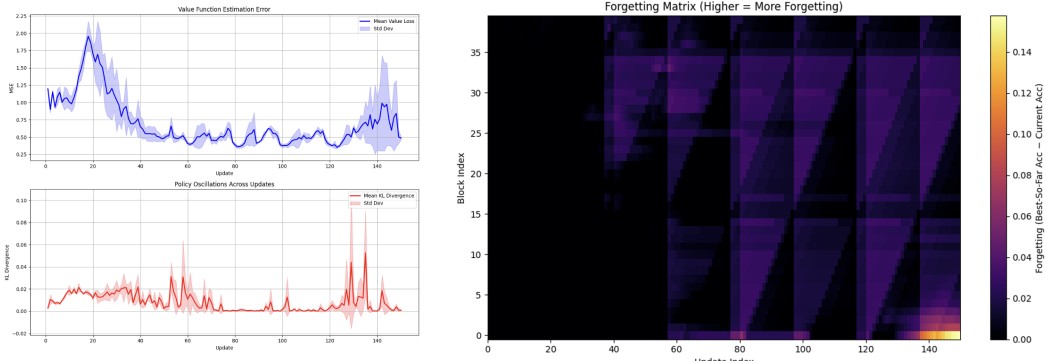

Figure 12: **Training is stable with Staggered Resets.** The left panel shows value function error (top) and policy KL divergence (bottom). Both metrics remain low and stable, with the value error (MSE) never exceeding 2.5. The right panel shows the forgetting matrix, which is almost completely dark, indicating that the agent successfully retains knowledge of all task stages throughout training.

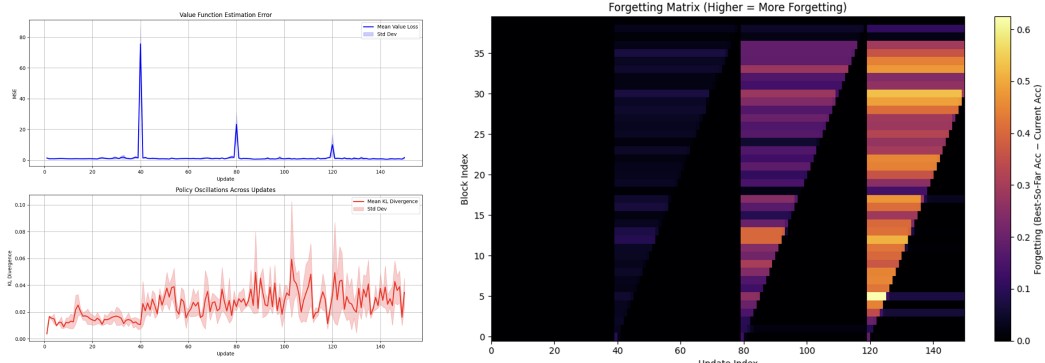

Figure 13: **Training is unstable with Synchronous (Naive) Resets.** The left panel shows catastrophic spikes in value function error (top, MSE > 80) that coincide with mass reset events (updates 40, 80, 120), as well as more erratic policy updates (bottom). The forgetting matrix on the right shows bright, cyclical patterns, indicating severe forgetting of early-stage skills as the training data shifts to later stages.

**Value Function Instability.** The value function error, visualized in the top-left plot of each figure, demonstrates the most dramatic effect. With synchronous resets (Figure 13), the value function experiences catastrophic prediction error spikes. The Mean Squared Error (MSE) surges to over 80 at regular intervals. These spikes align perfectly with the mass environment reset cycle (every 40 updates, as $H/K = 200/5 = 40$), which occurs when the data distribution abruptly shifts from late-episode states back to initial states. This provides clear evidence that the cyclical data stream makes it impossible for the critic to maintain a consistent estimate of state values. In stark contrast, the agent trained with staggered resets (Figure 12) maintains a stable and low value estimation error, never exceeding an MSE of 2.5.

**Policy Oscillations.** The policy oscillation metric, measured by the KL divergence between consecutive policy updates (visualized in the bottom-left plot of each figure), also shows a clear difference. While both methods exhibit some variance, the policy under synchronous resets (Figure 13) becomes significantly more erratic after the first mass reset (update 40), exhibiting larger and more frequent spikes in KL divergence. This indicates that the learning algorithm is making large, corrective updates in response to the sudden distributional shift, rather than the smooth, incremental learning facilitated by the stationary data from staggered resets.

**Catastrophic Forgetting.** The forgetting matrices, shown in the right panel of each figure, visualize the agent's inability to retain knowledge. For the agent with synchronous resets (Figure 13), the matrix reveals a stark sawtooth pattern of forgetting. As the training data progresses to later blocks in an episode cycle, the agent's accuracy on early blocks (e.g., blocks 0-10) is almost completely lost (bright yellow/orange). When the mass reset occurs, the agent must then relearn these initial skills. Conversely, the forgetting matrix for the staggered resets agent (Figure 12) is almost entirely black, signifying near-perfect knowledge retention across all task stages throughout training. Quantitatively, accuracy with naive resets drops by an average of 0.21 relative to its peak, while with staggered resets, the average drop is only 0.015—a 14-fold improvement in knowledge retention. This directly validates our central hypothesis: the cyclical data stream forces the agent into a destructive learn-forget cycle, which staggered resets completely mitigates.

