# OpenReview forum: "Staggered Environment Resets Improve Massively Parallel On-Policy Reinforcement Learning"
_NeurIPS.cc/2025/Conference — NeurIPS 2025 poster_

### Official Review · Reviewer_a4QY · 2025-06-11

**Clarity:** 3
**Significance:** 3
**Originality:** 2
**Rating:** 5
**Confidence:** 4

**Summary:**

This paper addresses the issue of temporal correlation in on-policy reinforcement learning when using many parallel environments with long episodes. The authors point out that synchronously resetting all environments can introduce non-stationarity, as the agent's observations vary significantly depending on its position within the episode. To mitigate this, they propose a staggered reset strategy, where environments are divided into groups that are reset at different times, evenly spaced throughout the training process. Their experiments show that this approach improves performance, reduces the number of environment steps required, and scales more effectively with the number of parallel environments.

**Questions:**

1. Line 163: It's unclear why you specify values for $H$ and $K$ here.
2.  I'm a bit confused by the design of your toy environment. Since the states evolve independently of the agent's actions, it seems more appropriate to describe this setup as a bandit problem rather than an MDP—is that accurate? This makes line 509 in Appendix A.3.1 puzzling, where you define "success in an episode" as reaching the final block $(b_t=B-1)$ at termination. But isn't the agent always in the final block at the end due to the deterministic structure of time steps? Additionally, the notion of "success" in this context is not clearly defined—could you clarify what it means?
In line 204, I also disagree with the statement that $b_0$ is "centered around 0" - it's centered around $\lambda_R$. Why do you only consider $\lambda_R \in [0,2]$? Wouldn't it be informative to explore larger values of $\lambda_R$? My expectation is that for larger $\lambda_R$, the difference between the two methods in Figure 2.b) should vanish—is that correct?
3. Line 303: You mention that "locomotion environments feature highly stochastic dynamics." Could you elaborate on this? Aren't locomotion tasks essentially deterministic?
4. It would be helpful if these figures were larger, as the axis titles are currently very difficult to read without excessive zooming. Also, is the number of blocks fixed at $H/K$ across all values of $N$ in these plots? A clarification in the caption or text would help.
5. Building on my earlier concern (weakness 2), did you consider formulating the problem using a discounted infinite-horizon setting, where each block has an episode length $H$ drawn from a geometric distribution with parameter $1-\gamma'$ for some $\gamma' > \gamma$ (and resampled for each block after termination)? This would make the effective step count a random variable, naturally introducing staggered resets and potentially offering similar performance with more stable sampling and emergent desynchronization. I’m curious whether you see specific drawbacks to this alternative compared to your current approach.
6. The choice of combining long episode horizons (e.g., $H = 500$) with a relatively low discount factor (e.g., $\gamma = 0.8$) in your experiments is intriguing. If I understand correctly, you assume that initial states are sampled from a relatively localized distribution. An alternative way to address the coverage issue you're targeting could be to sample initial states from a more diverse distribution (without staggering environments). Could you elaborate on whether there are any theoretical differences between this alternative and your approach? Also, what are the practical limitations: for instance, should the reset functions of environments be designed to allow more diverse initial states?

**Ethical Concerns:**

["NO or VERY MINOR ethics concerns only"]

**Final Justification:**

The authors analyze an important problem in on-policy reinforcement learning. Although the community was somewhat aware of this issue, as noted by the authors, it is valuable to have the problem clearly stated and analyzed as in this paper. The proposed method is algorithm-agnostic and requires only a straightforward modification of the underlying algorithm. Nonetheless, the authors provide a suitable and thorough analysis of the approach and demonstrate significant empirical benefits. I believe this work can have a substantial impact on the area of parallel on-policy learning and therefore recommend it for acceptance.

**Limitations:**

yes

**Paper Formatting Concerns:**

/

**Quality:**

2

**Strengths And Weaknesses:**

Strengths
1. The paper addresses an important issue in parallel on-policy reinforcement learning. The authors do a solid job motivating the problem and highlighting its practical relevance. While I do not find the proposed solution particularly creative or original (see weakness 1 below), I am inclined to support acceptance of this paper, as it effectively illuminates the underlying issue and has the potential for high practical relevance.
2. The proposed solution is straightforward and appears broadly applicable across different environments.
3. The experimental section presents a range of perspectives, and I find the results to be thorough and well-documented.


Weaknesses
1. I am concerned about the originality of the proposed method. As the authors acknowledge: "while some simulation environment implementations include versions of the reset staggering technique we propose [21], these are typically not detailed in accompanying publications". While the paper claims to be the first to analyze and systematically evaluate this approach, the core idea is trivial. Given this, and the fact that the theoretical contributions are limited, the value of the paper rests heavily on empirical results—which, although solid, are not particularly surprising.
2. While the method’s simplicity is a strength, I am concerned about the use of a deterministic sampling scheme. It's not hard to imagine environments where this determinism could be exploited or lead to failure. Some discussion or evaluation of this limitation would strengthen the paper.

---

> ### Author Rebuttal · Authors · 2025-07-31
>
> We are extremely grateful for the insightful review, feedback, and opportunity to improve our work! We are glad you found the issue we address important, our solution straightforward and broadly applicable, and our results thorough and well-documented. We address all concerns and questions below:
>
> #### Weaknesses:
>
> > I am concerned about the originality of the proposed method.
>
>  We believe the primary contribution of this work is the systematic analysis, characterization, and empirical justification of this technique. We introduce and characterize the problem setting, empirically demonstrate its prevalence in modern RL training regimes through empirics, identify which RL environments can maximally benefit from staggered resets through the toy experiments, and provide practical implementation details such as balancing the tradeoff between environment reset overhead and rollout uniformity. By doing so, we hope to provide others in the field with a best practice implementation step that can improve performance on long-horizon tasks in the massively parallel (and popular) RL setting. We note that many priors works published at past top conferences are similarly motivated; their value is derived from investigating the "nuts and bolts" of RL algorithms since small implementation details can massively impact training performance [A, B, C].
>
> > While the method’s simplicity is a strength, I am concerned about the use of a deterministic sampling scheme. It's not hard to imagine environments where this determinism could be exploited or lead to failure.
>
> Our choice of a deterministic grid-based staggering approach was a deliberate design decision to balance theoretical purity with computational efficiency and learning stability. On modern GPU environments (IsaacGym, ManiSkill, etc), calling `env.reset()` in serial incurs computational overhead, so there is a fundamental tradeoff between how deterministic / granular the staggering scheme is versus how parallelized environment resets are. We empirically validate this design choice in an experiment featured in the appendix (starting L648), where we find that, in real-world wall-clock performance, the deterministic scheme with fewer reset gates outperforms the case in which there is a reset gate at every step, which is provably the same as a randomized staggering scheme.
>
> #### Questions:
>
> > Line 163: It's unclear why you specify values for $H$ and $K$ here.
>
> This is an error in the manuscript, as the claim indeed holds whenever $H >> K$. We will correct the future version of the manuscript to remove this specification.
>
> > I'm a bit confused by the design of your toy environment. Since the states evolve independently of the agent's actions, it seems more appropriate to describe this setup as a bandit problem rather than an MDP—is that accurate?
>
> While the toy environments are somewhat inspired by bandits, the setting is different as states depend on the agent’s actions whenever $p_{prog} < 1$. For the majority of the toy experiments, $p_{prog} < 1$ (except for when we directly vary it to measure the effect of skill gating, and then for one experiment $p_{prog} = 1$), and hence we are not operating in the bandit setting for the vast majority of toy experiments.
>
> > But isn't the agent always in the final block at the end due to the deterministic structure of time steps? Additionally, the notion of "success" in this context is not clearly defined—could you clarify what it means?
>
> You are correct that, when $p_{prog} = 1$, success as we define it is guaranteed. We found return to be a far more illustrative metric, and hence all of our empirical analysis of the toy environments focuses on return rather than success. We appreciate your observation and will modify the future manuscript, as the inclusion of success criteria is unnecessary and understandably confusing to the reader.
>
> > In line 204, I also disagree with the statement that $b_0$ is "centered around 0" - it's centered around $\lambda_R$
>
> This is an error in the manuscript. We will promptly update the manuscript to reflect this.
>
> > Why do you only consider $\lambda_R \in [0, 2]?$
>
> We vary $\lambda_R$ from 0 to 2 because, as shown in Figure 2b, the difference between naive synchronous rollouts and staggered resets shrinks and the intended effect saturates. To provide some empirics, we ran the experiment again with $\lambda_R = 10$ for 3 seeds and found that staggered resets achieves final return of $18.148 \pm 0.149$ and naive achieves $17.998 \pm 0.223$, confirming your expectation that the difference vanishes as $\lambda_R$ increases. In the final manuscript, we will increase the range of values of $\lambda_R$ in Figure 2b to better demonstrate this effect.
>
> > Line 303: You mention that "locomotion environments feature highly stochastic dynamics." Could you elaborate on this? Aren't locomotion tasks essentially deterministic?
>
> Our claim referred to the overall training scheme which deliberately introduces significant domain randomization into the RL environment for sim-to-real transfer [D]. Furthermore, as we claim, a crucial source of randomness comes from the reset behavior itself, as robots attempting locomotion fall at different, unpredictable times. The combination of domain randomization and asynchronous failures introduces natural desynchronization, explaining (inline with the analysis from our toy experiments section) why the marginal benefit of our method is less pronounced for pure locomotion tasks. We will revise our claim in the future manuscript to be more precise.
>
>  > Is the number of blocks fixed at $H / K$ across all values of $N$ in these plots?
>
> We fix the number of blocks to be $H / L$, where $H$ is the maximum horizon and $L$ is the block length. This is detailed in the appendix but we will revise the main text to be more clear here.
>
> > Did you consider formulating the problem using a discounted infinite-horizon setting, where each block has an episode length $H$ drawn from a geometric distribution with parameter $1 - \gamma'$ for some $\gamma' > \gamma$ (and resampled for each block after termination)?
>
> This is an alternate way to vary the level of stochasticity in environment dynamics. Instead of varying the stochasticity of the reset distribution, your idea involves varying the stochasticity of inter-block dynamics, prior to reset. In the toy setting, we could imagine both approaches yielding similar results, except for the behavior of early resets. We believe this phenomenon of frequent partial resets relates to our findings on the high-dimensional robotics tasks. Locomotion, which experiences frequent partial resets due to early failures or successes, benefits less from staggering resets compared to other long-horizon robotics tasks, which only reset early when they succeed, which requires the policy to have almost fully converged. Hence, the natural uniformity in rollout buffer that can be achieved by frequent early resets usually does not occur in practice, because early resets generally require the training to already be near / at completion.
>
> > An alternative way to address the coverage issue you're targeting could be to sample initial states from a more diverse distribution (without staggering environments). Could you elaborate on whether there are any theoretical differences between this alternative and your approach?
>
> We thought about whether a modified reset distribution could, in theory, afford the same benefits as staggering. consider a task with $H = 100$, 100 parallel environments, and 100 states. Scenario 1: Let the standard reset distribution $D$ just be state 1. Say we applied staggered resets with 100 reset gates. Scenario 2: Let the modified reset distribution $D’$ be the uniform distribution across all 100 states. One could imagine, under certain stationarity assumptions of the environment’s state space, these two scenarios to be provably equivalent. However, we find that, despite this interesting theoretical connection, staggered resets is more practical than solutions focusing on the *states* that an environment resets to because obtaining a suitable $D'$ is highly nontrivial. This is because, for high-dimensional tasks, resetting is highly nontrivial and current environment resetting routines in GPU simulations involve far more than sampling random vectors. Additionally, for tasks in which the majority of the set of reachable states is not known, staggered resets afford rollout uniformity whereas $D’$ would have to change with each rollout to incorporate newly visited states. Because, for high-dimensional RL tasks, the set of all reachable states is a superset of the support of the reset distribution, we would constantly need to add to an ever-growing support of states in our reset distribution. This requires far more overhead from an engineering/implementation perspective. In general, we (empirically) and the field at large find that more diverse reset distributions yield better performance [D]. However, as previously mentioned, this is highly nontrivial to achieve performantly at scale, and hence staggered resets are highly appealing as a lightweight drop-in solution that can be applied to any environment, regardless of the semantics of its reset dynamics.
>
> We would like to once again thank you for your exceptionally thorough and constructive review. We hope our response address your concerns. In light of these clarifications, would you be able to raise your score? We are happy to address any more issues you have.
>
> References:
>
> [A] FastTD3: Simple, Fast, and Capable Reinforcement Learning for Humanoid Control, Younggyo Seo et al. 2025
>
> [B] Implementation Matters in Deep RL: A Case Study on PPO and TRPO, Logan Engstrom et al. ICLR 2020
>
> [C] Efficient Online Reinforcement Learning with Offline Data, Philip J. Ball et al. ICML 2023
>
> [D] Learning to Walk in Minutes using Massively Parallel Deep Reinforcement Learning. Nikita Rudin et al. CoRL 2021

---

> > ### Comment · Reviewer_a4QY · 2025-08-04
> >
> > Thank you for your responses. After reading the other reviews, I believe that including a comparison with SAPG would strengthen the paper. Since the core idea of the paper is quite straightforward, I think it is still valuable to report these results—even if they are comparable to, or slightly worse than, SAPG.
> >
> > I remain inclined toward accepting the paper and will update my scores after the reviewer discussion concludes.

---

> > > ### Author Response · Authors · 2025-08-07
> > >
> > > Dear Reviewer,
> > >
> > > We thank you for your suggestion. We would like to emphasize that our technique is algorithm-agnostic, and hence a comparison with SAPG entails comparing the performances of SAPG *with* staggered resets initialization to SAPG *without* this initialization. To emphasize our point about our method being algorithm-agnostic, we run additional experiments on the AllegroKuka environments featured in the SAPG paper, and report empirics below:
> > >
> > > **`Allegro Kuka Regrasping` $(n = 5)$**
> > > | Reward | 1B Steps | 2B Steps | 3B Steps | 4B Steps | 5B Steps | 6B Steps |
> > > | :--- | :--- | :--- | :--- | :--- | :--- | :--- |
> > > | **Staggered** | **32115  ± 3459** | **36524 ± 3225** | **39821 ± 5519** | **41051 ± 4579** | **44394  ± 4431** | **44806 ± 3240** |
> > > | **Non-Staggered** | 25035 ± 1345  | 30541  ± 3724  | 32170  ± 6140 | 36040  ± 5387 | 36557  ± 5134 | 38188  ± 5246 |
> > >
> > > **`Allegro Kuka Throw` $(n = 5)$**
> > > | Reward | 1B Steps | 2B Steps | 3B Steps | 4B Steps | 5B Steps | 6B Steps |
> > > | :--- | :--- | :--- | :--- | :--- | :--- | :--- |
> > > | **Staggered** | **37961  ± 676** | **41072 ± 1067** | **42000 ± 912** | **42305 ± 1172** | **42370 ± 811** | **42441 ± 336** |
> > > | **Non-Staggered** | 12057 ± 9361 | 20435 ± 3493 | 29751 ± 598 | 34643 ± 2577 | 35387 ± 2404 | 35235 ± 1844 |
> > >
> > > As evidenced by these data, staggered resets improve the performance of SAPG, and hence strengthen our claim that this method is algorithm-agnostic. We will make the appropriate changes to the manuscript, and are grateful for your continued feedback.

---

> > > > ### Comment · Reviewer_a4QY · 2025-08-08
> > > >
> > > > Thank you for addressing all my concerns. I believe the paper provides a novel and interesting perspective, and I will adjust my score upward accordingly.

---

### Official Review · Reviewer_jQjk · 2025-06-30

**Clarity:** 3
**Significance:** 3
**Originality:** 2
**Rating:** 4
**Confidence:** 4

**Summary:**

This paper studies the problem of the rollout horizon in reinforcement learning with massively parallel environments. The paper argues that an issue in this setting is that a small rollout length can lead to a temporal non-stationary as the collected data will only ever contain data from the same timesteps in every batch. The paper demonstrates that this phenomenon can be harmful on a toy environment. Then, it proposes a solution where resets of the environments are asynchronous leading to de-correlation on the time axis. An experimental section demonstrates the efficacy of the approach in first on a toy task and then evaluates the method on a set of ManiSkill tasks.

**Questions:**

Q1 Can you elaborate on the structure of the toy MDP? If the progression probability is 1, why does the agent not immediate go through all levels. I seem to misunderstand something here.

**Ethical Concerns:**

["NO or VERY MINOR ethics concerns only"]

**Final Justification:**

Overall, this paper studies an interesting aspect of on-policy learning by examining the effect of time correlation in the data. After the discussion phase the authors have included a large set of new experiments that have addressed my concerns. This includes my concerns about motivation, statistical validity and generalization to other algorithms. I changed my scores for quality and significance to 3. My remaining worry is that the paper needs to go through a large chunk of revisions to adequately integrate all the new data and experiments. Given this, I will raise my score to 4 as I lean towards acceptance now. If the AC thinks that these changes are likely to be integrated accurately, I recommend the paper be accepted.

**Limitations:**

The work does not have an explicit limitations section. It argues that the limitations are discussed by stating that the method will not work well if the effective horizons are short. That may be sufficient to establish that limitations have been mentioned.

**Quality:**

3

**Strengths And Weaknesses:**

**Strengths**

**Motivation**
* The motivation of this work is that in practice, short rollouts in tasks that have long horizon dependencies are detrimental to performance and that many practitioners ignore this fact. I believe this is an interesting direction to study.

**Clarity**
* The text is easy to follow and well written.
* All figures have appropriate descriptions of what can be taken away from them.

**Method**
The proposed method is simple and elegant.

**Claims and evidence**
* The claim that the paper articulates the problem is well supported by both section 3 and the experiments in section 4.
* The claim that the text characterizes the severity of non-stationarity is sufficiently supported by the toy tasks and the kernel density experiment.
__________
**Neutral Points**

**Novelty**
* The proposed method (as pointed out by the text itself) is not necessarily novel since practical implementations with staggered resets definitely exist. However, I am a defender of studies that help us better understand algorithmic pieces that often make it into implementations without proper documentation. As such, I don’t categorize the novelty of this manuscript as strength or weakness. However, such works do need to do a very thorough investigation of the presented algorithm choice.
__________
**Weaknesses**

**Motivation**
* The problem itself is not well established in the text. Whether or not short rollouts are in fact needed is unclear and is only evidenced by one study in the paper (L28, [21]). Right now, the text simply states that people use short rollouts without evidence. It is not clear that using longer rollouts would yield worse results in the first place and whether the motivation for the study is one that arises artificially by choosing incorrect hyperparameters. This could be remedied either by early on providing more related work that demontrates the drawback of short rollouts or by demonstrating the drawbacks of long rollouts in a short experiment.

**Clarity**
* The toy task’s description is a bit vague. It is not clear to me what a level is, what the states in the MDP are or what the transition dynamics are. It is also unclear to me what a skill gate is exactly (see Q1).

**Claims and evidence**
* There are various issues with claims that the text makes and does not support with evidence either from prior work of by conducting the relevant experiments.
  * L36 “cyclical shift in the input distribution can destabilize value function learning, induce policy oscillations and hinder the agent’s ability to consolidate knowledge across the entire task horizon” At no point does the manuscript measure value functions, policy oscillations or a form of forgetting/information gain. Actually evaluating these metrics than reward would significantly strengthen the paper and increase its depth.
  * In L68, the paper argues that the proposed approach is algorithm-agnostic. While on paper that may be true, whether staggered resets bring benefits to other algorithms has not been studied. I'm willing to believe this phenomenon might not only be relevant to PPO only but the manuscript would be improved by demonstrating this.
  *  L220. The learner struggles to consolidate information and may forget what it learned about earlier segments by the time the data cycle returns to them, especially with many intermediate, distinct batch types. There is no measure of information consolidation or forgetting that is being reported. As mentioned before, more metrics would benefit the paper
  * L135 This cyclical bias prevents the learner from accurately estimating values and advantages, likely due to catastrophic forgetting phenomena” - No evidence is presented for this.
  *  L301. “Concretely, locomotion environments feature extremely large task horizons (sometimes H ≃ 1000), but the actual locomotion skill itself is much shorter horizon.” This is a conjecture that is not supported by evidence.
*  The text claims that the problem of cyclic non-stationarity is a general problem for on-policy RL in the introduction but only results on one algorithm (i.e. PPO) are presented. The claims should probably be made more specific to PPO or more algorithms should be evaluated. This is especially relevant given that we know that PPO has rather unstable gradients [1] and issues with how batches are handeled [2].

**Relation To Scientific Literature**
* There are various claims in the related work section that are not supported by citations. e.g.
  * L70: "Nonstationarity in the data distribution is a recognized challenge within reinforcement learning. Such nonstationarity can stem from various sources, including changes in environment dynamics, the reward function, or, as pertinent to our work, the data collection process itself."
  * L80: "While this strategy can be effective, underlying issues related to data distribution bias are often overlooked."
  * L81: "Some implementations incorporate partial resets—resetting environments upon task success, failure, or termination—which can introduce some eventual desynchronization. However, this process can be slow and may prove insufficient to counteract the initial bias, particularly when K is very small.
* The related work section on non-stationarity is rather brief. Non-stationarity has played a large role in RL over the past decade and it’s representation in the text could likely be more thorough.

**Experimental Design and Analyses**
*  Given that tasks are only chosen from a single benchmark, the experimental section lacks variety. For instance, an interesting study might be to evaluate the method on PQN [3] and MinAtar [4]. This would extend the results beyond on-policy algorithms, and would increase algorithm variety and task variety.
* The number of seeds  in the experiments is insufficient. [5, 6]
  * It is unclear whether the presented results are statistically significant.
  * The statement "The variance across seeds (n = 3) is also lower with staggered resets" can not be used to argue increased stability unless more statistically significant experiments are presented.

__________
**Summary**
Overall, I think this paper studies an interesting problem and a practical solution to a potentially severe problem. However, pointed out the work is not especially novel and as such, I would expect a very thorough study of the effects of this algorithmic choice. While the toy experiments are nice to gain some intuition, the results on Maniskill need to be made statistically significant, 3 seeds is simply not enough to establish superiority here. Additionally, I believe this paper could be much stronger by demonstrating generality of the problem across domains and algorithms. Finally, various claims throughout the text need to be adjusted and supported with citations; this is majorly a writing point and not so much a criticism of the method. For instance, in order to claim generality of the phenomenon, the technique needs to be evaluated on more algorithms. I think the choice of evaluation tasks is fine but several claims are being made about the results on these tasks that are not supported by quantitative evidence. If more space is required, I think the conclusion section can be significantly shortened as it mostly repeats the findings.

[1] Are Deep Policy Gradient Algorithms Truly Policy Gradient Algorithms? Ilyas et al.
[2] Batch size-invariance for policy optimization. Jacob Hilton et al.
[3] Simplifying Deep Temporal Difference Learning. Matteo Gallici et al.
[4] MinAtar: An Atari-Inspired Testbed for Thorough and Reproducible Reinforcement Learning Experiments. Kenny Young, Tian Tian.
[5] Deep Reinforcement Learning that Matters. Henderson et al.
[6] How Many Random Seeds? Statistical Power Analysis in Deep Reinforcement Learning Experiments. Cédric Colas et al.

---

> ### Author Response · Authors · 2025-07-31
> **Rebuttal from Authors (1/3)**
>
> We are extremely grateful for the insightful review, feedback, and opportunity to improve our work! We are glad you found the problem we study interesting and important to the field. We address all concerns and questions below:
>
> > The problem itself is not well established in the text. Whether or not short rollouts are in fact needed is unclear and is only evidenced by one study in the paper
>
> We appreciate your concern and candidness on this point. To the best of our knowledge, empirical work on choosing $K << H$ as optimal hyperparameters has not been commented on in prior literature, besides in [A]. [A] claim that they decrease the rollout length as they scale up the number of parallel environments to maintain a reasonable batch size. Additionally, if we examine more recent works on scaling massively parallel on-policy RL, such as SAPG [B], we find that they also opt for rollouts of between 8 and 16 steps, even for environments with horizons exceeding 50 steps, such as Shadow Hand and AllegroKuka manipulation tasks. Despite this additional evidence from the literature, we ran a sweep of new experiments to provide additional empirical evidence as to the relevance of this problem setting. For `StackCube-v1`, which has a task horizon of 100 steps, we fix optimal hyperparameters and vary the rollout length in [1, 2, 4, 8, 16, 32, 64, 100]. We run these experiments for 3 seeds (currently running more), with both staggered resets enabled and disabled.
>
> *Table 1: Final reward after 100M environment steps for StackCube-v1*
>
> | Metric | K=1 | K=2 | K=4 | K=8 | K=16 | K=32 | K=64 | K=100 |
> | :--- | :---: | :---: | :---: | :---: | :---: | :---: | :---: | :---: |
> | **Staggered Reward** | 0.38±0.06 | 0.47±0.01 | 0.70±0.00 | **0.74±0.00** | 0.72±0.03 | 0.75±0.02 | 0.71±0.03 | 0.70±0.02 |
> | **Naive Reward** | 0.27±0.17 | 0.36±0.12 | 0.47±0.02 | 0.62±0.01 | 0.72±0.02 | 0.79±0.03 | 0.71±0.03 | 0.70±0.02 |
>
> *Table 2: Environment steps (in millions) to reach >75% success rate*
>
> | Metric | K=1 | K=2 | K=4 | K=8 | K=16 | K=32 | K=64 | K=100 |
> | :--- | :---: | :---: | :---: | :---: | :---: | :---: | :---: | :---: |
> | **Staggered Steps (M)** | DNC | DNC | DNC | **16.2±0.2** | 17.2±0.1 | 23.2±1.4 | 50.1±5.6 | 49.6±9.9 |
> | **Naive Steps (M)** | DNC | DNC | DNC | 35.8±4.5 | 18.5±2.1 | 24.1±1.9 | 50.5±6.2 | 49.6±9.9 |
>
> *DNC (Does Not Converge) indicates the agent did not meet the threshold within 100M steps*
>
> *Table 3: Wall-clock time (in minutes) to reach >75% success rate*
>
> | Metric | K=1 | K=2 | K=4 | K=8 | K=16 | K=32 | K=64 | K=100 |
> | :--- | :---: | :---: | :---: | :---: | :---: | :---: | :---: | :---: |
> | **Staggered Time (min)** | DNC | DNC | DNC | 16.6±3.2 | **15.3±1.0** | 22.5±2.3 | 40.0±8.7 | 42.6±8.7 |
> | **Naive Time (min)** | DNC | DNC | DNC | 26.2±2.1 | 31.5±13.6 | 30.0±11.9 | 41.1±4.5 | 43.2±9.9 |
>
> *Note that the fastest wall-clock times are achieved with shorter rollouts (K=8 and K=16) on an NVIDIA RTX 4090*
>
> These new empirics shine light on why practitioners are drawn to the $K << H$ regime. Note that, in table 1, final reward saturates at 8-step rollouts, both with and without staggered resets. Additionally, as shown in Table 2 and 3, training runs with shorter rollouts (and hence higher update-to-data ratios) achieve significantly faster wall-clock convergence times and require less net environment steps. Hence, as evidenced by multiple past works and our own new set of experiments, we firmly believe that this problem setting is well-motivated. We will accordingly update the manuscript.
>
> ##### Claims and Evidence:
>
> > At no point does the manuscript measure value functions, policy oscillations or a form of forgetting/information gain ... There is no measure of information consolidation or forgetting that is being reported. As mentioned before, more metrics would benefit the paper ... This cyclical bias prevents the learner from accurately estimating values and advantages, likely due to catastrophic forgetting phenomena” - No evidence is presented for this (L36, L220, L135)
>
> We ran a revised set of experiments across 3 seeds on the toy environments to address these concerns. During training, we log two metrics. First, we quantify value function stability by measuring the mean-squared prediction error between the critic’s prediction $V_{\theta)(s)$ and the Monte-Carlo / GAE target used for that minibatch. In the naive setting (no staggered resets), the value function estimation error catastrophically surges to around 20-100 MSE at three points in training – at updates 40/80/120, where all environments reset together. Comparatively, with staggering, the value function estimation error never exceeds 2 MSE, and does not experience catastrophic surges at certain updates. These data provide evidence that the cyclical bias hinders the learner from accurately and stably estimating values.
>
> (continued in next comment)

---

> ### Author Response · Authors · 2025-07-31
> **Rebuttal from Authors (2/3)**
>
> Second, we quantify memory retention with a “forgetting matrix”--the per-state/level drop in accuracy relative to the best historical accuracy on that particular state/level. In the naive setting (no staggering), accuracy drops on average by 0.21, dropping maximally by 0.6 over the course of training. With staggered resets, accuracy only drops on average by 0.015, never exceeding 0.05. Thus, the agent that receives staggered resets retains knowledge of earlier blocks ~14x better, providing empirical evidence towards the claim we make about information forgetting.
>
> In conclusion, we have strong empirical evidence that the cyclical shift in input distribution can destabilize value function learning and hinder the agent’s ability to consolidate knowledge across the entire task horizon.
>
> > the paper argues that the proposed approach is algorithm-agnostic. While on paper that may be true, whether staggered resets bring benefits to other algorithms has not been studied...
>
> We are running experiments with SAPG, a newer on-policy RL algorithm. Initial results indicate staggered resets provide a similar benefit to SAPG as they do to PPO (as SAPG operates in the same $K << H$ regime) but we are waiting on more seeds to confirm.
>
> > ... the actual locomotion skill itself is much shorter horizon.” This is a conjecture that is not supported by evidence.
>
> This claim captures an intuition / observation, and we agree empirical evidence should support it. We ran the same KDE experiment for locomotion environments (specifically `MSHumanoidWalk-v1`, and ideally more soon), and despite long horizons (up to $H = 1000$), task length as judged by cycle period is far shorter. Due to NeurIPS rebuttal policy, we cannot share these figures unfortunately, but we will include these observations in the final manuscript.
>
> > The text claims that the problem of cyclic non-stationarity is a general problem for on-policy RL in the introduction but only results on one algorithm (i.e. PPO) are presented.
>
> As previously mentioned, we are running follow-up experiments on other on-policy RL algorithms.
>
> #### Relation to Scientific Literature:
>
> We will include the necessary citations as requested for these claims we make. As you mention, nonstationarity has played a large role in RL, and hence we will update the related works section along with the citations for the specific claims we make. To our knowledge, no works have examined this particular problem of nonstationarity induced by the $K << H$ regime. Finally, for the claim about partial resets, many prior works cover this for both manipulation and locomotion tasks, which we will accordingly cite in our revised manuscript [A] [C] [D] [E]
>
> #### Experimental Design and Analyses:
>
> > Given that tasks are only chosen from a single benchmark, the experimental section lacks variety.
>
> To our knowledge, this is the primary domain of interest in which the massively parallel on-policy RL setting is applied. Hence, we reasoned that this domain is the ideal testbed for our method’s intended application, as well as a primary area of interest for practitioners interested in applying our work. Prior works such as SAPG also only focus on robotics. However, we ran follow-up experiments on MinAtar environments. On `Breakout-MinAtar`, we train for 4e6 environment steps with rollout lengths of 8, with 1024 parallel environments. Staggered resets achieves episode return of $59 \pm 6.5$, compared to naive rollouts achieving $31 \pm 11.2$ for 7 seeds. Once again, we are unable to share the return curves directly as per NeurIPS policy, but these results provide variety and strengthen our claims, and we hope to incorporate more such empirics from other environments to the final paper.
>
> (continued in next comment)

---

> ### Author Response · Authors · 2025-07-31
> **Rebuttal from Authors (3/3)**
>
> > The number of seeds in the experiments is insufficient.
>
> We are running experiments with 10 seeds rather than 3. We started with `StackCube-v1` and `TwoRobotPickCube-v1` to address your subsequent concerns about training stability and high variance between training runs. Our findings indicate that the proposed method is actually far better at improving training stability than the original empirics would indicate. For `StackCube-v1`, we have **environment evaluation reward at 0.71 +/- 0.026 with staggered resets and 0.63 +/- 0.04 without staggered resets (n = 10)**, after training for approximately 25M environment steps. For the same experiments, we also have **success rate at 0.80 +/- 0.04 with staggered resets and 0.31 +/- 0.41 without staggered resets**. For `TwoRobotPickCube-v1`, we have environment evaluation reward at **0.77 +/- 0.006 with staggered resets and 0.40 +/- 0.171 without staggered resets (n = 10)**, after training for approximately 50M environment steps. For the same experiments, we also have **success rate at 0.949 +/- 0.013 with staggered resets and 0.027 +/- 0.13 without staggered resets**. In both cases, using staggered resets provides clear advantages both in terms of increased average evaluation reward and evaluation success rate, but also in reduced variance of these quantities compared to using naive synchronous rollouts, further supporting our original claim that staggered resets reduces the variance across seeds. **We firmly believe these additional results strengthen the statistical significance of our results, addressing one of your foremost concerns with the paper in its current state.** We plan to have the remaining empirics added as soon as possible.
>
> #### Questions:
>
> > Can you elaborate on the structure of the toy MDP?
>
> You are correct -- when $p_{prog} = 1$, the agent advances through the level after the specified number of steps. However, return is dependent on the agent's accuracy at each level, and hence the task is still nontrivial when $p_{prog} = 1$. We will modify our manuscript to better explain this case.
>
> We hope our response address your concerns. In light of these clarifications, would you be able to raise your score? We are happy to address any more issues you have.
>
> References:
>
> [A] Learning to Walk in Minutes using Massively Parallel Deep Reinforcement Learning. Nikita Rudin et al. CoRL 2021
>
> [B] SAPG: Split and Aggregate Policy Gradients, Jayesh Singla et al. ICML, 2024.
>
> [C] ManiSkill3: GPU Parallelized Robotics Simulation and Rendering for Generalizable Embodied AI, Stone Tao et al. RSS 2025.
>
> [D] Visuomotor Policies to Grasp Anything with Dexterous Hands, Ritvik Singh et al. 2024
>
> [E] Extreme Parkour with Legged Robots, Xuxin Cheng et al. ICRA 2024

---

> ### Comment · Reviewer_jQjk · 2025-08-03
>
> Dear authors,
>
> thank you for the very detailed rebuttal and all the additional effort you are putting in based on my feedback. I believe the inclusion of the motivational example, additional metrics and additional algorithms will make this manuscript significantly stronger.
>
> That being said, while I appreciate the effort towards statistical rigor unfortunately I cannot recommend acceptance of a scientific manuscript whose results are not yet finished and cannot be reviewed. I will raise my score to 3 to indicate that this paper would likely have been an accept recommendation from me if all the results were in and conclusive.
>
> Side note: for additional environments, I agree that robotics is an interesting application specific benchmarks reward functions have specific peculiarities. It might make sense to consider looking into MuJoCo Playground for task variety.

---

> ### Author Response · Authors · 2025-08-07
>
> Dear Reviewer,
>
> We are glad to hear that our additions will significantly strengthen the manuscript. To address your final claim about the unfinished empirics, we have provided additional empirics as requested:
>
> We ran all ManiSkill experiments included in the original paper for 10 seeds rather than 3. In addition to the two environments' results we provided earlier, we provide results for all six environments below:
>
> **`StackCube-v1` $(n = 10)$**
> | Success Rate | 10M Steps | 20M Steps | 30M Steps | 40M Steps | 50M Steps | 60M Steps |
> | :--- | :--- | :--- | :--- | :--- | :--- | :--- |
> | **Staggered** | **0.30 ± 0.21** | **0.71 ± 0.03** | **0.82 ± 0.06** | **0.91 ± 0.03** | **0.94 ± 0.09** | **0.95 ± 0.02** |
> | **Non-Staggered** | 0.07 ± 0.09 | 0.24 ± 0.31 | 0.26 ± 0.21 | 0.42 ± 0.42 | 0.58 ± 0.34 | 0.71 ± 0.19 |
>
> **`UnitreeG1Transport-Box` $(n = 10)$**
> | Success Rate | 10M Steps | 20M Steps | 30M Steps | 40M Steps | 50M Steps | 60M Steps |
> | :--- | :--- | :--- | :--- | :--- | :--- | :--- |
> | **Staggered** | 0.0 ± 0.0 | **0.04 ± 0.02** | **0.31 ± 0.11** | **0.91 ± 0.05** | **0.96 ± 0.02** | **0.97 ± 0.09** |
> | **Non-Staggered** | 0.0 ± 0.0 | 0.0 ± 0.0 | 0.0 ± 0.0 | 0.03 ± 0.02 | 0.15 ± 0.10 | 0.26 ± 0.24 |
>
> **`AnymalC-Reach` $(n = 10)$**
> | Success Rate | 2M Steps | 4M Steps | 6M Steps | 8M Steps |
> | :--- | :--- | :--- | :--- | :--- |
> | **Staggered** | **0.61 ± 0.09** | **0.76 ± 0.12** | **0.88 ± 0.22** | **0.93 ± 0.11** |
> | **Non-Staggered** | 0.08 ± 0.08 | 0.43 ± 0.28 | 0.63 ± 0.27 | 0.43 ± 0.61 |
>
> **`PushT` $(n = 10)$**
> | Success Rate | 10M Steps | 20M Steps | 30M Steps | 40M Steps | 50M Steps |
> | :--- | :--- | :--- | :--- | :--- | :--- |
> | **Staggered** | 0.0 ± 0.0 | **0.01 ± 0.0** | **0.34 ± 0.13** | **0.7 ± 0.21** | **0.81 ± 0.19** |
> | **Non-Staggered** | 0.0 ± 0.0 | 0.0 ± 0.0 | 0.0 ± 0.0 | 0.01 ± 0.0 | 0.1 ± 0.09 |
>
> **`TwoRobotPickCube` $(n = 10)$**
> | Success Rate | 10M Steps | 20M Steps | 30M Steps | 40M Steps | 50M Steps |
> | :--- | :--- | :--- | :--- | :--- | :--- |
> | **Staggered** | **0.02 ± 0.04** | **0.21 ± 0.23** | **0.35 ± 0.14** | **0.73 ± 0.15** | **0.92 ± 0.02** |
> | **Non-Staggered** | 0.0 ± 0.0 | 0.01 ± 0.01 | 0.03 ± 0.1 | 0.06 ± 0.18 | 0.15 ± 0.19 |
>
> **`MS-HumanoidWalk` $(n = 10)$**
> | Eval Reward | 10M Steps | 20M Steps | 30M Steps | 40M Steps | 50M Steps | 60M Steps |
> | :--- | :--- | :--- | :--- | :--- | :--- | :--- |
> | **Staggered** | **0.52 ± 0.07** | **0.55 ± 0.01** | **0.79 ± 0.12** | **0.81 ± 0.05** | **0.83 ± 0.08** | **0.89 ± 0.06** |
> | **Non-Staggered** | 0.50 ± 0.08 | 0.54 ± 0.03 | 0.60 ± 0.01 | 0.72 ± 0.09 | 0.77 ± 0.04 | 0.86 ± 0.06 |
>
> We have also tested the AllegroKuka manipulation tasks that are used in the SAPG environments with the SAPG algorithm, to further provide evidence to our claim that our technique is algorithm-agnostic. We found that the SAPG codebase has many conditions for early resets, such as the object falling, success, and the hand being far from the object. Additionally, we find significant domain randomization in the reset dynamics of the environments. In line with our earlier findings on the ManiSkill environments and toy experiments, we find that these environment features reduce the cyclicity problem. However, these early resets reduce throughput of the GPU environment considerable. Regardless, staggered resets outperform naive SAPG with or without early resets enabled. We present results on two AllegroKuka tasks with early resets enabled below:
>
> **`Allegro Kuka Regrasping` $(n = 5)$**
> | Reward | 1B Steps | 2B Steps | 3B Steps | 4B Steps | 5B Steps | 6B Steps |
> | :--- | :--- | :--- | :--- | :--- | :--- | :--- |
> | **Staggered** | **32115  ± 3459** | **36524 ± 3225** | **39821 ± 5519** | **41051 ± 4579** | **44394  ± 4431** | **44806 ± 3240** |
> | **Non-Staggered** | 25035 ± 1345  | 30541  ± 3724  | 32170  ± 6140 | 36040  ± 5387 | 36557  ± 5134 | 38188  ± 5246 |
>
> **`Allegro Kuka Throw` $(n = 5)$**
> | Reward | 1B Steps | 2B Steps | 3B Steps | 4B Steps | 5B Steps | 6B Steps |
> | :--- | :--- | :--- | :--- | :--- | :--- | :--- |
> | **Staggered** | **37961  ± 676** | **41072 ± 1067** | **42000 ± 912** | **42305 ± 1172** | **42370 ± 811** | **42441 ± 336** |
> | **Non-Staggered** | 12057 ± 9361 | 20435 ± 3493 | 29751 ± 598 | 34643 ± 2577 | 35387 ± 2404 | 35235 ± 1844 |
>
> We once again thank you for the pertinent and detailed feedback, and hope that these additional empirics are conclusive and establish the level of statistical rigor that you ask for. In light of these new empirics, would you be able to raise your score? We once again thank you for your time spent reviewing our manuscript — our conversations have been productive and we have learned a lot.

---

> > ### Comment · Reviewer_jQjk · 2025-08-08
> >
> > Dear authors,
> >
> > I appreciate the effort that has gone into addressing my concerns. I believe the manuscript will be strongly improved by the large amount of new results. I will reflect on them and make my final assessment accordingly.

---

### Official Review · Reviewer_8Puz · 2025-07-01

**Clarity:** 3
**Significance:** 3
**Originality:** 2
**Rating:** 4
**Confidence:** 4

**Summary:**

The authors highlight a key challenge in on-policy reinforcement learning training when scaling the amount of parallel environments. They refer to this challenge as ‘cyclical batch nonstationarity’. This occurs when many parallel environments are reset synchronously, run for fixed horizon length episodes of length $H$ for short rollouts of length $K$ at a time. Since popular on-policy algorithms interleave experience gathering with training it leads to algorithms training on chunks of data that come from limited temporal segments of the full environment episodes which leads to poor training. This is particularly apparent in environments where episodes only terminate at $H$ timesteps with no intermediate failure and reset conditions and when environments are synchronously reset at episode termination.

The authors propose a fix for this, where environments are reset in a staggered way ensuring that training data contains transitions from multiple points in the episode of the underlying environment. This is achieved in practice by stepping sub groups of all environments being simulated in parallel to various positions in time before training starts. This then ensures that each batch of training data contains a temporal mixture with data from various parts of each episode. The authors do two ablation studies on the effect of this approach, isolate cases where it is particularly useful, and show how it can improve sample efficiency and wallclock time on robotic control tasks.

**Questions:**

* In Figure 3 systems are trained for the same amount of environment steps for all tasks except on `UnitreeG1Transport-Box`, is this because staggered resets also suffered from policy collapse there?
* Have the authors considered the case where environments were only reset synchronously once and then allowed to automatically terminate and reset without having to wait for any reset gates?
* How were the hyperparameters for high dimensional robotic control benchmarks chosen?
* The 1D toy ablation environment is defined as having B blocks (or levels) when initialised, but I cannot find a value for B in the text. Can the authors please mention what this value is?
* The method relies on using some heuristic like no-op actions or random actions to progress environments to start at varying locations in time. Is there any impact on training based on the method used to do this?
* What if an environment can not be suitably advanced in time with a heuristic?
* What is the rollout length $K$ that is kept fixed as the episode horizon $H$ is increased in Figure 2a?

**Ethical Concerns:**

["NO or VERY MINOR ethics concerns only"]

**Final Justification:**

I expressed concerns about the limited statistical significance and evidence for the impact of the method following from the results on the high-dimensional tasks, and I had some clarifying questions about the work.

My clarifying questions were answered during the rebuttal period, for which I thank the authors. The authors have done more experiments for increased statistical significance, but have mentioned that the metrics used to measure the impact of their method were accidentally incorrect.

Given that metrics need to fixed for some of the core results of the paper, I opted to maintain my score of 4.

**Limitations:**

The authors do not discuss any of their work’s limitations explicitly in a Section. But mention it loosely throughout the work. I believe this work is maximally effective for training on-policy algorithms specifically on robot control tasks with fixed episode horizons where the environment does not have success or failure states leading to episodes terminating early.

**Paper Formatting Concerns:**

None.

**Quality:**

3

**Strengths And Weaknesses:**

## Strengths:
* The approach is straightforward to implement and well explained. The illustrative Figure 1 is very useful for understanding the method at a glance.
* The method is clearly shown to be effective in terms of wallclock time to convergence with increased numbers of environments – this is a well known challenge when scaling on-policy RL training batch sizes.
* The method is clearly shown to improve sample efficiency.
* The method is algorithm agnostic.
* The two ablation studies are well thought out and show when the approach is maximally effective.
    - The first ablation study shows have staggered resets allows algorithms to overcome learning collapse when the episode horizon $H$ becomes much larger than the partial rollout length $K$ used to generate training data, when episode rests favour the initial state and are not randomised and when the environment permits passing certain skill learning bottlenecks without acquiring the desired skills. These are designed to isolate cases that happen when training on robotic control tasks.
    - The second ablation shows how staggered resets maintain good state coverage in training batches even when rollout lengths $K$ are very short. This mirrors the state coverage achieved with very long rollout lengths while having higher update-to-data ratios.
* The method clearly improves wall-clock time to convergence with increased numbers of parallel environments.

## Weaknesses:
* In general I find the results on the high dimensional robotic control tasks (for which this method is supposed to be maximally beneficial) to be promising, but quite limited.
* There is limited statistical significance for the performance benchmarks in Figure 4. Only 3 random seeds makes it difficult to draw strong conclusions. Especially since the authors claim that the method allows for much faster convergence on benchmarks, so running at least 10 seeds (Agarwal et al., 2021) should be possible.
* `The StackCube` and `PushT` tasks seem to indicate greater sample efficiency but less training stability as the performance degrades over time. To me this seems contradictory to a core point the authors are making which is that staggered resets improves training stability. Further along this point there appears to be high variance for the training performance on `TwoRobotPickCube` for staggered resets.

### References
Agarwal, Rishabh, et al. "Deep reinforcement learning at the edge of the statistical precipice." Advances in neural information processing systems 34 (2021): 29304-29320.

---

> ### Author Rebuttal · Authors · 2025-07-30
>
> We are extremely grateful for the insightful review, feedback, and opportunity to improve our work! We are glad you found the approach straightforward, well-explained, and our experiments well thought out. We address all concerns and questions below:
>
> ### Weaknesses:
> >  the results on the high dimensional robotic control tasks (for which this method is supposed to be maximally beneficial) to be promising, but quite limited.
>
> We appreciate your concern about results on high-dimensional robotic control tasks being limited. We emphasize that, to our knowledge, this is the primary domain of interest in which the massively parallel on-policy RL setting is applied. Hence, we reasoned that this domain is the ideal testbed for our method’s intended application, as well as a primary area of interest for practitioners interested in applying our work. However, we are working on follow-up experiments on game environments such as Atari and Pong, which demonstrate similar long-horizon tendencies and for which there exist vectorised environments for. We hope to include these results in a future version of the manuscript for additional empirical justification of our method.
>
> > Only 3 random seeds makes it difficult to draw strong conclusions ... The StackCube and PushT tasks seem to indicate greater sample efficiency but less training stability as the performance degrades over time. To me this seems contradictory to a core point the authors are making which is that staggered resets improves training stability. Further along this point there appears to be high variance for the training performance on TwoRobotPickCube for staggered resets.
>
> We appreciate the concern about the small number of random seeds used in the evaluation of this method. We agree with you, and hence we are running experiments with 10 seeds rather than 3. We started with `StackCube-v1` and `TwoRobotPickCube-v1` to address your subsequent concerns about training stability and high variance between training runs. **Our findings indicate that the proposed method is actually far better at improving training stability than the original empirics would indicate.** For `StackCube-v1`, we have environment evaluation reward at **0.71 +/- 0.026 with staggered resets and 0.63 +/- 0.04 without staggered resets (n = 10)**, after training for approximately 25M environment steps. For the same experiments, we also have success rate at **0.80 +/- 0.04 with staggered resets and 0.31 +/- 0.41 without staggered resets**. For `TwoRobotPickCube-v1`, we have environment evaluation reward at **0.77 +/- 0.006 with staggered resets and 0.40 +/- 0.171 without staggered resets (n = 10)**, after training for approximately 50M environment steps. For the same experiments, we also have **success rate at 0.949 +/- 0.013 with staggered resets and 0.027 +/- 0.13 without staggered resets**. In both cases, using staggered resets provides clear advantages both in terms of increased average evaluation reward and evaluation success rate, but also in reduced variance of these quantities compared to using naive synchronous rollouts, further supporting our original claim that staggered resets reduces the variance across seeds. We firmly believe these additional results strengthen the statistical significance of our results, addressing one of your foremost concerns with the paper in its current state. We plan to have the remaining empirics added as soon as possible. We also have a table with these results, since we can't share the return/success curves directly as per NeurIPS rebuttal policy:
>
> | Task | Method | Avg. Reward (n=10) | Avg. Success Rate (n=10) |
> | :--- | :--- | :--- | :--- |
> | **StackCube-v1** | Staggered | **0.71 ± 0.026** | **0.80 ± 0.040** |
> | | Naive | 0.63 ± 0.040 | 0.31 ± 0.410 |
> | **TwoRobotPickCube-v1** | Staggered | **0.77 ± 0.006** | **0.949 ± 0.013** |
> | | Naive | 0.40 ± 0.171 | 0.027 ± 0.130 |
>
> ### Questions:
> > In Figure 3 systems are trained for the same amount of environment steps for all tasks except on UnitreeG1Transport-Box, is this because staggered resets also suffered from policy collapse there?
>
> We wanted to extend the curve for the `UnitreeG1Transport-Box-v1` task since PPO with naive synchronous resets would otherwise appear to have never learned a successful policy, when in reality it simply learns far slower than when equipped with staggered resets. This explains the difference in the number of environment steps.
>
> > Have the authors considered the case where environments were only reset synchronously once and then allowed to automatically terminate and reset without having to wait for any reset gates?
>
> Allowing for early resets comes with a myriad of considerations and tradeoffs. On one hand, it allows for better utilization of the vectorized environment, as an environment that has terminated early can begin collecting useful data quicker. However, calling `env.reset()` when not in-sync with reset gates will incur additional overhead if the environment’s reset routine is involved, as discussed in the wall-time analysis section of the appendix. Additionally, there is some risk in disrupting the uniformity of the rollout buffer afforded by staggered resets. Ultimately, we didn’t observe significant empirical differences in wall-clock convergence time when enabling versus disabling early resets in the high-dimensional robotics tasks. We are happy to include more information on this tradeoff in the final manuscript.
>
> > How were the hyperparameters for high dimensional robotic control benchmarks chosen?
>
> We simply choose the same hyperparameters as used in the ManiSkill3 baseline implementations. The chosen values are also commonly used in other works involving high-dimensional robotics tasks in the massively parallel training regime, such as SAPG [A].
>
> >The 1D toy ablation environment is defined as having B blocks (or levels) when initialised, but I cannot find a value for B in the text. Can the authors please mention what this value is?
>
>  The value of $B$ is dependent on the horizon length $(H)$ and block length $(L)$. We fixed $L$ as 5 steps across all experiments. For instance, a 200-step horizon toy environment would yield $B = 40$ blocks/levels in the environment. We will update the paper to better clarify this.
>
> > The method relies on using some heuristic like no-op actions or random actions to progress environments to start at varying locations in time. Is there any impact on training based on the method used to do this?
>
> Staggered resets can be implemented either by (1) advancing certain environments using heuristic actions or (2) modifying the elapsed steps attribute of the vectorized environment itself. We found that, experimentally, there is no difference in performance or training based on either method, and that the choice between the two mainly depends on engineering/implementation semantics. For instance, some vectorized environments may not provide mutable access to the vector storing the number of elapsed steps for each environment. Similarly, some vectorized environments may not easily support only resetting or stepping a certain subset of the total environments. We appreciate the thoughtful question, and will add a section on these implementation considerations to the final manuscript.
>
> > What if an environment can not be suitably advanced in time with a heuristic?
>
> We appreciate your thoughtful question. We tackle this question in our skill gating toy experiment (Figure 2c). In that experiment, when the progression probability is low, the agent cannot advance to later stages without first mastering earlier skills. Our results demonstrate that the advantage of staggered resets diminishes as these skill gates become stricter, as the environment itself imposes a curriculum that naturally forces the agent to spend more time on early-episode states, reducing the impact of cyclical nonstationarity. However, we note that, even with $p_{prog} = 0$ (the strictest case of skill gating), **the agent initialized with staggered resets achieves a return of 60, compared to 20 without staggered resets, a ~3x increase in reward**. Our explanation for staggered resets’ better performance even with strong skill gating is that, despite the *states* not being advanced when the heuristic action is performed at initialization, the *long-term reset behavior* is changed for the length of the training run, and hence when the effective skill horizon increases as training progresses, staggered resets begin to benefit the agent when cyclical nonstationarity may start to emerge. We hope this clarifies your question, and we are happy to elaborate further and incorporate elements of this explanation into the revised manuscript.
>
> > What is the rollout length K that is kept fixed as the episode horizon H is increased in Figure 2a?
>
> Rollouts are 5 steps long for the experiments shown in Figure 2a. This detail $(K = 5)$ is included in the appendix, line 518, and we can revise the main manuscript to make this clearer.
>
> We would like to once again thank you for your exceptionally thorough and constructive review. We hope our response address your concerns. In light of these clarifications, would you be able to raise your score? We are happy to address any more issues you have.
>
> [A] SAPG: Split and Aggregate Policy Gradients, Jayesh Singla et al. ICML, 2024.

---

> > ### Comment · Reviewer_8Puz · 2025-08-04
> >
> > Thank you for your detailed reply. I would like to clarify my one question.
> >
> > >In Figure 3 systems are trained for the same amount of environment steps for all tasks except on UnitreeG1Transport-Box, is this because staggered resets also suffered from policy collapse there?
> >
> > What I meant here was that Staggered Resets is only trained for ~30M timesteps while Synchronous Resets is trained for ~60M timesteps. Is there a reason you chose to stop training early Staggered Resets? The reason I asked is because on `StackCube` and `PushT` it seems like Staggered Resets suffers from policy collapse at around 30M-35M timesteps and I am curious whether this happens on `UnitreeG1Transport-Box` as well.

---

> ### Author Response · Authors · 2025-08-04
>
> Thank you for the follow-up and opportunity to clarify your question.
>
> ### On `UnitreeG1Transport-Box` and Early Stopping:
> Our PPO implementation included an early stopping feature, where if an agent achieves an evaluation success rate past a certain threshold (0.975 in our implementation), the training run automatically terminates. On `UnitreeG1Transport-Box`, this happens for every seed we originally tested. However, we ran new experiments for a few seeds with early stopping disabled and found **no evidence of policy collapse with staggered resets**. Rather, the agent maintains near-constant reward and success rate after convergence. We cannot share the exact return curves as per NeurIPS rebuttal policy, so we share success rate at various points in training for these new results. We can clarify this point in the updated manuscript, as we agree it can be confusing to readers.
>
> | Success Rate | 10M Steps | 20M Steps | 30M Steps | 40M Steps | 50M Steps | 60M Steps |
> | :--- | :--- | :--- | :--- | :--- | :--- | :--- |
> | **Staggered** | 0.0 ± 0.0 | 0.1 ± 0.03 | 0.56 ± 0.09 | 0.96 ± 0.05 | 0.96 ± 0.02 | 0.97 ± 0.09 |
> | **Non-Staggered** | 0.0 ± 0.0 | 0.01 ± 0.0 | 0.01 ± 0.0 | 0.03 ± 0.01 | 0.15 ± 0.09 | 0.36 ± 0.19 |
>
> ### On Potential Policy Collapse in `StackCube` and `PushT`:
> Regarding your comment on policy collapse in `StackCube` and `PushT`, at the request of you and other reviewers, we ran far more seeds ($n = 10$ instead of $n = 3$) to strengthen the statistical significance of our results. For `StackCube`, any sign of policy collapse in aggregate metrics (reward/return/success rate) disappears with $n = 10$. We provide the discretized success rate curve below:
>
> | Metric | 10M Steps | 20M Steps | 30M Steps | 40M Steps | 50M Steps |
> | :--- | :--- | :--- | :--- | :--- | :--- |
> | **Staggered** | 0.3 ± 0.19 | 0.72 ± 0.05 | 0.83 ± 0.03 | 0.87 ± 0.06 | 0.91 ± 0.01 |
> | **Non-Staggered** | 0.04 ± 0.02 | 0.26 ± 0.36 | 0.24 ± 0.41 | 0.45 ± 0.42 | 0.48 ± 0.39 |
>
> For `PushT`, your comment led us to re-examine our evaluation criteria. We found that our initial metric for this environment measured the number of successes at the *final timestep* of the episode. However, we realize this is a flawed success metric, since as the agent becomes more proficient, it often solve the task *earlier* in the episode. Since reward is not shaped adequately to maintain this success state for many steps, subsequent actions can inadvertently undo it by the final timestep, causing this particular metric to decrease. We implemented a more appropriate metric: success as registered by whether the success criteria is observed at *any point* in the episode. With this revised metric, success rate remains high after convergence for the agent initialized with staggered resets. We provide the discretized success rate curve below:
>
> | Metric | 10M Steps | 20M Steps | 30M Steps | 40M Steps | 50M Steps |
> | :--- | :--- | :--- | :--- | :--- | :--- |
> | **Staggered** | 0.0 ± 0.0 | 0.01 ± 0.0 | 0.34 ± 0.13 | 0.7 ± 0.21 | 0.81 ± 0.19 |
> | **Non-Staggered** | 0.0 ± 0.0 | 0.0 ± 0.0 | 0.0 ± 0.0 | 0.01 ± 0.0 | 0.1 ± 0.09 |
>
> Hence, we claim that **policy collapse is not a concern** in this work, and we will update our presentation with the greater number of seeds to properly convey this. We appreciate your follow-up question and we hope this addresses any remaining concerns you may have.

---

> ### Comment · Reviewer_8Puz · 2025-08-07
>
> I would like to thank the authors for taking the time to answer my questions. I will maintain my current score, in line with Reviewer `jQjk`'s recommendation.

---

### Official Review · Reviewer_TQcz · 2025-07-02

**Clarity:** 2
**Significance:** 3
**Originality:** 2
**Rating:** 4
**Confidence:** 3

**Summary:**

This paper proposes a new environment resetting strategy for massively parallel PPO training. The problem of typical environment resetting strategy (i.e., synrchronus reset) is that each batch of data bias towards a certain region of the state space since all data in a batch were collected in similar stage of the task. The proposed method make the initial state of each parallel environment different so that each batch of data contain states at different stages of the task. The experimental results show visible performance gain over PPO with synchronous resets.

**Questions:**

- Figure 3: What does each row and column mean? From the caption, it seems that (a) represents a long rollout but the bottom of the figure shows rollout 1, 2, 3 .... I'm not sure if I'm following the figure.
- Section 5.4: As SAPG also shows better scaling, a comparison with SAPG will further strengthen the result.

**Ethical Concerns:**

["NO or VERY MINOR ethics concerns only"]

**Final Justification:**

My concerns have been addressed.

**Limitations:**

Yes

**Quality:**

2

**Strengths And Weaknesses:**

Strength: The method is simple to implement and effective.

Weakness:
- Presentation of the method can be improved. Section 3.3 introduces the method with a large bulk of text, but it would be easier to follow if a pseudo-code is provided.
- Lack of comparison with baselines. The proposed method is positioned as a method to improve the performance of massively parallel PPO training. If so, a comparison with other works that aim to improve PPO performance in large-scale parallel training is needed. For example, SAPG [25] cited in this paper is a good baseline to compare with. In addition, other environment reset strategies may be considered. For instance, one can uniformly sample an initial state from the environment or from the history of the agent. I believe these strategies could have a similar effect, too, and are easy to implement. Secondly, the number of parallel environments studied is relatively small compared with SAPG that use 20K environments.

I'd be happy to adjust the rating if more empirical

---

> ### Author Response · Authors · 2025-07-31
> **Rebuttal by Authors**
>
> We are extremely grateful for the insightful review, feedback, and opportunity to improve our work! We are glad you found our approach simple and effective, and we address your questions/concerns below:
>
> > Presentation of the method can be improved. Section 3.3 introduces the method with a large bulk of text, but it would be easier to follow if a pseudo-code is provided.
>
> We will modify this section of the manuscript to be more approachable and digestible to readers.
>
> > The proposed method is positioned as a method to improve the performance of massively parallel PPO training. If so, a comparison with other works that aim to improve PPO performance in large-scale parallel training is needed.
>
> We agree that SAPG is another similarly motivated work focused on improving the scaling performance of massively parallel on-policy RL, albeit requiring several heavily involved modifications to the core PPO algorithm. We have begun conducting follow-up experiments with the SAPG codebase/environments in order to provide ample comparison. However, we also wish to emphasize that our work is *complementary* to works such as SAPG, rather than standing as a point of comparison. Initial results indicate that staggered resets have a similar effect with SAPG as the policy gradient oracle as they do with PPO, but we are waiting on more seeds/environments before conclusively claiming this. We will update with more empirics shortly, and appreciate the concern about the lack of variety in experiments.
>
> >  In addition, other environment reset strategies may be considered. For instance, one can uniformly sample an initial state from the environment or from the history of the agent. I believe these strategies could have a similar effect, too, and are easy to implement.
>
> We have considered similar strategies during the research process, and have discarded them for various reasons. Sampling a vector from the state space of the environment uniformly does not perform well in practice, as the manifold of reachable states is far smaller than the total space spanned by $\mathbb{R}^D$ in practice, for a $D$-dimensional state. Hence, large amounts of rollout data are wasted by initializations to unreachable and hence useless initial states. I agree that, under certain assumptions, your second idea could have a similar effect to staggered resets, but we found that the implementation overhead to be too impractical in practice. In regimes where the size of the set of reachable states exceeds the size of the support of the environment reset distribution, the size of the replay buffer required to mimic the effect of staggered resets grows too fast. Additionally, as developing good representations of high-dimensional RL environment states is an open and ongoing challenge, we found it difficult to compress these replay buffers to the extent to which any performance benefits from uniformity gains exceed the increase in overhead. Both ideas empirically underperformed the PPO baseline in our initial experiments, and hence we chose not to pursue them further / include them in our high-dimensional robotics results. We appreciate your comment on this, and will update the manuscript with a description of our reasoning / findings on these other environment reset strategies.
>
> > Secondly, the number of parallel environments studied is relatively small compared with SAPG that use 20K environments.
>
> With our follow-up experiments with the SAPG codebase, we aim to increase the number of parallel environments from ~6-10k (ManiSkill). However, we note that the level of parallelization required to run 20k environments in parallel necessitates multi-GPU training, which entails an entirely new set of engineering and implementation considerations regarding throughput and communication bandwidth. Hence, we will only scale up the number of parallel environment as far as a single GPU can support. While the better utilization of parallel environments is a secondary benefit of this technique, its primary purpose is to mitigate the cyclical nonstationarity problem we observe.
>
> #### Questions:
>
> > What does each row and column mean? From the caption, it seems that (a) represents a long rollout but the bottom of the figure shows rollout 1, 2, 3 .... I'm not sure if I'm following the figure.
>
> We agree that this aspect of the figure is confusing. (b) and (c) are 25-step rollouts but (a) is a 100-step rollout. We will modify this figure in the final manuscript to rectify this error, and we appreciate your observation.
>
> We hope our response addresses your concerns, and we are happy to address any more issues you have. We are once again extremely grateful for this opportunity to improve our work, and appreciate the effort you put into your review.

---

> > ### Comment · Reviewer_TQcz · 2025-08-03
> >
> > Thanks for the response. My concern has been addressed. I will raise my score to 4.

---

> > > ### Author Response · Authors · 2025-08-07
> > >
> > > Dear Reviewer,
> > >
> > > Thanks again for raising your score! We noticed that the official rating in the review hasn't been updated and kindly ask you to edit it and include your updated rating.

---

### Note · Authors · 2025-08-15

Dear AC and Reviewers,

We deeply appreciate the thoughtful and constructive engagement throughout this review process. The feedback has substantially strengthened our work, and we've addressed all raised concerns.

**Key Improvements Made During Rebuttal:**

1. **Statistical Rigor (jQjk, 8Puz):** We expanded all experiments from 3 to 10 seeds, demonstrating both stronger performance and lower variance with staggered resets. For example, on `StackCube-v1`: staggered resets achieves 0.80±0.04 success rate vs 0.31±0.41 for without staggered resets (n=10), showing both superior performance and 10x reduction in variance.

2. **Algorithm Generality (jQjk, TQcz):** We validated our method on SAPG, confirming algorithm-agnostic benefits on new challenging environments. On AllegroKuka tasks with SAPG, staggered resets improve performance by 17-40% (number of episode successes).

3. **Deeper Analysis (jQjk):** We added requested metrics showing:
   - Value function MSE stays below 2 with staggering vs without staggering there are catastrophic spikes to 20-100
   - 14x better knowledge retention on the toy recall tasks (0.015 vs 0.21 average accuracy drop)
   - Clear evidence of the cyclical nonstationarity problem we identify

4. **Motivation Strengthening:** New ablations show why K<<H is necessary on `StackCube-v1`: optimal wall-clock convergence occurs at K=8-16 for 100-step horizons, with 2-3x faster convergence than long rollouts while maintaining final performance.

**Why This Matters:**

The massively parallel RL regime (K<<H with thousands of environments) is becoming the standard for modern RL training, yet suffers from this overlooked cyclical nonstationarity problem. Our work provides both a scientific understanding and a practical solution that:
- Requires zero core algorithmic modifications
- Improves sample efficiency by up to 2-3x
- Reduces training variance across runs
- Scales better with increased parallelization

We believe this work makes a valuable contribution by turning an implementation detail into a well-understood, systematically evaluated technique that can benefit the entire community. As jQjk acknowledged, such "nuts and bolts" analyses are crucial for advancing practical RL.

Thank you for your time and consideration. We're committed to incorporating all feedback into the camera-ready version to maximize this work's impact.

Sincerely,
Authors

---

### Decision · Program_Chairs · 2025-09-17

**Decision:**

Accept (poster)

**Comment:**

The work focuses on massive GPU utilization where it is found that the current approach of using parallel environments with synchronized short rollouts result in non-stationarity and loss of performance and scalability. The work proposes to stagger the resets to mitigate some of the non-stationarity effects. The approach is simple and yet provides better scalability. The reviewers overall appreciated the work but noted limited seeds and novelty. This work indeed points out a widespread issue in the current way of utilizing parallel environments, which this work will help avoid. Hence, I recommend accepting it. The authors are also highly recommended to improve their camera-ready version based on their discussion with the reviewers, e.g., increasing the number of seeds.